



# Comparison of different Aethalometer correction schemes and a reference multi-wavelength absorption technique for ambient aerosol data

Jorge Saturno[1], Christopher Pöhlker[1], Dario Massabò[2], Joel Brito[3], Samara Carbone[4], Yafang Cheng[1],
Xuguang Chi[5], Florian Ditas[1], Isabella Hrabě de Angelis[1], Daniel Morán-Zuloaga[1], Mira L. Pöhlker[1],
Luciana V. Rizzo[6], David Walter[1], Qiaoqiao Wang[1], Paulo Artaxo[7], Paolo Prati[2] and Meinrat O.
Andreae[1,8,9]

[1]Max Planck Institute for Chemistry, Biogeochemistry and Multiphase Chemistry Departments, P. O. Box 3060, 55020 Mainz, Germany
[2]Department of Physics & INFN, University of Genoa, via Dodecaneso 33, 16146, Genova, Italy
[3]Laboratory for Meteorological Physics, University Blaise Pascal, Clermont-Ferrand, France
[4]Institute of Agrarian Sciences, Federal University of Uberlândia, Uberlândia, Minas Gerais, Brazil
[5]Institute for Climate and Global Change and School of Atmospheric Sciences, Nanjing University, China
[6]Department of Earth and Exact Sciences, Institute of Environmental, Chemical and Pharmaceutics Sciences, Federal University of São Paulo, São Paulo, Brazil
[7]Department of Applied Physics, Institute of Physics, University of São Paulo, Rua do Matão, Travessa R, 187, CEP 05508-900, São Paulo, SP, Brazil
[8]Scripps Institution of Oceanography, University of California San Diego, La Jolla, CA 92098, USA
[9]Geology and Geophysics Department, King Saud University, Riyadh, Saudi Arabia

*Correspondence to*: Jorge Saturno (j.saturno@mpic.de)

**Abstract.** Deriving absorption coefficients from Aethalometer attenuation data requires different corrections to compensate for artifacts related to filter-loading effects, scattering by filter fibers, and scattering by aerosol particles. In this study, two different correction schemes were applied to 7-wavelength Aethalometer data, using Multi-Angle Absorption Photometer (MAAP) data as a reference absorption measurement at 637 nm. The compensation algorithms were compared to 5-wavelength offline absorption measurements obtained with a Multi-Wavelength Absorbance Analyzer (MWAA), which serves as a multiple-wavelength reference measurement. The online measurements took place in the Amazon rainforest, from the wet-to-dry transition season to the dry season (June – September 2014). The mean absorption coefficient (at 637 nm) during this period was $1.8 \pm 2.1$ Mm$^{-1}$, with a maximum of 15.9 Mm$^{-1}$. Under these conditions, the filter-loading compensation was negligible. One of the correction schemes was found to artificially increase the short-wavelength absorption coefficients. It





was found that accounting for the aerosol optical properties in the scattering compensation significantly affects the absorption Ångström exponent (AAE) retrievals. Proper Aethalometer data compensation

schemes are crucial to retrieve the correct AAE, which is commonly implemented in brown carbon contribution calculations. We found that a 'hybrid' algorithm was more appropriate to achieve optimal correlations with the MAAP absorption coefficients and with the AAE retrieved from offline MWAA measurements.

## 1 Introduction

Aerosol particles scatter and absorb solar radiation in the atmosphere and thus have an important impact on the Earth's radiative budget and climate (Andreae and Ramanathan, 2013; IPCC, 2013; Penner et al., 1992; Yu et al., 2006). Light absorption by atmospheric aerosols is dominated by black carbon (BC), an aerosol species that is emitted by incomplete combustion of biomass or fossil fuels (Bond and Bergstrom, 2006). Black carbon absorbs radiation from infrared to near-UV wavelengths and leads to

positive radiative forcing (IPCC, 2013). Other light absorbing aerosols include a class of organics called *brown carbon* (BrC) (Andreae and Gelencsér, 2006), mineral dust (Myhre and Stordal, 2001), and *primary biological aerosol particles* (PBAP) (Després et al., 2012). High uncertainties still remain regarding the aerosol interactions with solar radiation (Andreae and Ramanathan, 2013; Bond et al., 2013), especially because ambient aerosol absorption is often measured over a limited wavelength range

or at only one wavelength.

The wavelength dependence of aerosol light absorption is expressed by the absorption Ångström exponent (AAE) (Ångström, 1929). The AAE of fresh fossil-fuel derived BC is typically around 1.0, i.e., the absorption changes as a function of $\lambda^{-1}$ (Bergstrom et al., 2002). However, when BC particle size is larger than 50 nm or becomes coated by non-absorbing materials, the AAE can decrease below 1.0

(Lack and Langridge, 2013). Moreover, the bulk aerosol wavelength dependence can significantly increase in the presence of other light absorbers, such as BrC (Andreae and Gelencsér, 2006; Kirchstetter, 2004), reaching high values between 3.5 and 7.0. Assuming a fixed spectral dependence of 1 for BC, several studies have estimated the BrC contribution as a function of AAE (Favez et al., 2010;



Sandradewi et al., 2008). However, given the uncertainties associated to the AAE of BC, these methods

could potentially provide erroneous BrC estimations (Garg et al., 2016; Wang et al., 2016).

Absorption coefficients and BC mass concentrations are related by the mass absorption cross-section

(MAC) (Bond et al., 2013). Ground-based continuous measurements of BC mass concentrations and

absorption coefficients are required to retrieve the appropriate ambient aerosol MAC values, since this

relationship and its wavelength dependence are affected by the by the mixing state and physical and

chemical conditions of the aerosol particles (Flowers et al., 2010; Lack and Cappa, 2010; Moosmüller et

al., 2011). Moreover, retrieving the wavelength dependence of ambient aerosol requires absorption

measurements at two or more wavelengths.

Only few commercially-available techniques offer multi-wavelength absorption measurements. The

most commonly used methods are filter-based techniques, including a modified version of the Particle

Soot Absorption Photometer (PSAP) (Virkkula et al., 2005), which measures at three different

wavelengths in the visible spectral region, and the Aethalometer (Hansen et al., 1984), which measures

the attenuation of light at two or seven different wavelengths (2-λ vs. 7-λ instrument). The above

mentioned instruments are filter-based techniques that determine attenuation and suffer from various

artifacts (detailed discussion below), converting attenuation coefficients to absorption coefficients

requires several corrections (Arnott et al., 2005; Collaud Coen et al., 2010; Schmid et al., 2006;

Virkkula et al., 2007; Weingartner et al., 2003) that generally need concomitant scattering and additional

absorption measurements. The correction process of filter-based measurement artifacts introduces

uncertainties in the AAE that are difficult to determine (Collaud Coen et al., 2010).

Two well known artifacts affect filter-based absorption measurements by enhancing or reducing the

effective optical path length. One of them is related to the multiple scattering effects, which induces a

positive bias of light attenuation. The multiple scattering effects are caused by the *scattering by the*

*filter fibers* and the *scattering by aerosol particles on the filter*. The scattering by aerosol particles

depends on the optical properties and size distribution of the measured aerosol particles. On the other

hand, the second effect is related to the "*shadowing*" that deposited aerosol particles cause on each

other. This effect, called *filter-loading effect*, reduces the optical path length in the filter and depends on

the amount and optical properties of the deposited particles.





The bias related to multiple scattering effects can be reduced by measuring the radiation reflected by the filter at different angles and simulating the radiation transfer. This principle was incorporated in the design of the Multi-Angle Absorption Photometry (MAAP) technique (Petzold and Schönlinner, 2004).

The design consists of a single-wavelength instrument (637 nm LED light source) that measures the transmitted radiation through a glass-fiber filter and the reflectance at two different angles (130° and 165°). Using this configuration and a radiation transfer model, the instrument is able to account for the mentioned artifacts related to multiple scattering and provides approximately "corrected" absorption coefficients (Petzold et al., 2005).

For accurate estimation of absorption coefficients and their spectral dependency, Aethalometer measurements rely on a number of correction procedures, a compilation of different correction schemes can be found in Collaud Coen et al., (2010). The first systematic correction scheme that deals with the different artifacts affecting Aethalometer measurements was proposed by Weingartner et al. (2003). This correction scheme uses a comparison with an indirect light absorption measurement (extinction minus

scattering) to estimate a multiple scattering compensation. In addition, a filter-loading correction was estimated by empirically calculating a "shadowing factor". This correction consists of the following empirical equation that converts attenuation coefficients, $\sigma_{ATN}$, into absorption coefficients, $\sigma_{ap}$,

$$\sigma_{ap} = \frac{\sigma_{ATN}}{C \cdot R(ATN)} \qquad (1)$$

where C accounts for multiple scattering effects on the filter, due to: a) scattering by the filter fibers and

b) scattering by aerosol particles embedded on the filter. The factor R(ATN) accounts for the filter-loading effect.

Later, Virkkula et al. (2007) proposed another filter-loading correction through calculating the average attenuation before and after a filter change. This correction applied a compensation factor in the form of $(1 + k *ATN)$, where $k$ is calculated for each filter change and ATN corresponds to the attenuation. A

similar approach was used to design the dual-spot technology Aethalometer (model AE33) that intrinsically compensates for filter-loading effects using a two beam system with different flow rates (Drinovec et al., 2015).





In a detailed study, Arnott et al. (2005) introduced a scattering correction factor that accounts for the aerosol particle scattering artifact. In a similar way, Schmid et al. (2006) proposed a correction

algorithm that included a parameterization of the scattering by filter fibers and scattering by aerosol particles as a function of AAE and an iteration procedure to obtain corrected absorption coefficients. Both correction schemes used PAS measurements at 532 nm as a reference absorption measurement. Later, by using MAAP absorption measurements as a reference, Collaud Coen et al. (2010) evaluated the above-mentioned correction algorithms and proposed two new ones based on the Schmid and Arnott

corrections. Their algorithms, among several changes to the previous ones, included a new scattering correction parameterization that uses measured optical properties of the aerosol particles instead of the "standard" ones implemented in Schmid and Arnott correction algorithms. The comparison made by Collaud Coen et al. (2010) resulted in a good agreement between MAAP and Aethalometer BC measurements when using the "Schmid-like" correction algorithm. On the other hand, the "Arnott-like"

algorithm lead to many negative $\sigma_{ap}$ values, especially under low absorption conditions (Collaud Coen et al., 2010).

Previous studies on Aethalometer compensation schemes have evaluated corrected absorption coefficients in comparison to reference absorption measurements (PAS or MAAP), which were done at only one wavelength. In this study, we use a Multi-Wavelength Absorbance Analyzer (MWAA),

introduced by Massabò et al. (2013, 2015), to conduct a systematic multi-wavelength evaluation of ambient data. This way we can estimate the impact of the most common and reliable Aethalometer correction schemes on the AAE uncertainties. We used collected MAAP filter samples from long-term aerosol measurements in central Amazonia to perform offline multi-wavelength absorption measurements using the MWAA. The results presented here are relevant for the study of valuable multi-

wavelength data provided by the widely used Aethalometers.



## 2 Materials and methods

### 2.1 Sampling site and selected data period

Field measurements were carried out at the Amazon Tall Tower Observatory (ATTO) (S 02° 08.602'; W 59° 00.033'), located in the Uatumã Sustainable Development Reserve, Amazonas State, Brazil, in the
central Amazon Basin. The site is located 150 km NE of the city of Manaus, upwind of the urban plume. A detailed description of the site can be found in Andreae et al. (2015).

The atmospheric aerosol was collected by using a 60 m 1-inch diameter stainless steel inlet tube without size cut-off, installed on a triangular mast since early 2014. The laminar flow rate in the inlet was constant at 30 lpm. The aerosol stream relative humidity was decreased down to 30 – 40 % by using
diffusion driers. In this study, we corrected the data for standard temperature and pressure (273.15 K and 1013.25 hPa) and did not apply any correction to compensate for particle losses. The sampling period analyzed here comprises the wet-to-dry transition time (June – July 2014) and part of the dry season (August – September 2014). In the beginning of the measurement period (beginning of June), aerosol particle number concentrations were very low, in the order of 100-400 cm$^{-3}$, measured by a
Condensation Particle Counter (CPC) (Andreae et al., 2015). These typical wet season conditions slightly changed during the transition season until the end of August when particle number concentrations increased to around 500-2000 cm$^{-3}$ (Andreae et al., 2015). The selected measurement period was a good opportunity to evaluate the Aethalometer performance under different conditions. During this period, the aerosol absorption coefficients increase from near detection limit values to the
highest values measured at the ATTO site during the dry season.

### 2.2 Instrumentation

A 7-λ Aethalometer (model AE31, Magee Scientific Company, Berkeley, USA), nominal wavelengths: 370, 470, 520, 590, 660, 880, and 950 nm, was used to measure attenuation coefficients $\sigma_{ATN}$, which are reported by the instrument as BC mass concentrations. Details about the measurement principle and the
different corrections to the data are explained in the next section.

Scattering coefficients, $\sigma_{sp}$, were measured by a 3-λ nephelometer (Model Aurora 3000, Ecotech Pty Ltd., Knoxfield, Australia), nominal wavelengths: 450, 525, and 635 nm. The instrument was manually





calibrated using $CO_2$ as span gas. Zero tests and spans were conducted periodically. The scattering coefficients measured by the instrument were corrected for truncation errors following the method

proposed by Müller et al. (2011), using the sub-µm correction factors as function of the scattering Ångström exponents. The detection limits, calculated as three standard deviations of 1-min resolution particle-free air measurements, were 1.1, 0.9, and 0.7 Mm$^{-1}$ at 450, 525, and 635 nm, respectively. Due to a malfunction of the 635 nm channel, we excluded those data from our calculations.

A Multi-Angle Absorption Photometer, MAAP (Model Carusso 5012, Thermo Electron Group,

Waltham, USA) was used to measure the absorption coefficient at 637 nm. The instrument uses a glass-fiber filter tape, where the aerosol particles are collected on a sample spot. Light transmission (at 0°) and reflectance at two different angles (130° and 165°) are measured every 5 min (Petzold et al., 2005). A radiative model calculation provides the light absorption coefficient derived from the absorbance measurements and accounts for the light scattering by filter fibers and aerosol particles deposited on the

filter. The instrument reports BC mass concentrations calculated by assuming a mass absorption cross-section (MAC) of 6.6 m² g$^{-1}$, based on Bond et al. (2006). A measurement bias after every filter change can occur if the absorption coefficients exceed ~20 Mm$^{-1}$ (Hyvärinen et al., 2013), which was not the case during the period of this study. The instrument sampled at a flow rate of 500 l h$^{-1}$ in series with the nephelometer and was configured to trigger a filter change when transmission reached a minimum of

60% or after 24 h. Therefore, more samples were collected during the dry season, when the aerosol particle concentration was higher and the transmission threshold was reached quickly. All data obtained from the online measurements (Nephelometer, Aethalometer, and MAAP) were aggregated to 30-min means. MAAP data below the detection limit (0.132 Mm$^{-1}$ with 30-min resolution) were excluded from the analysis.

The MWAA was used to measure the light absorption coefficients on MAAP-collected filter samples. This instrument was developed by Massabò et al. (2013) and measures the light transmitted through a filter sample (forward hemisphere) and the light reflected at two different angles (backward hemisphere) in a similar configuration to the MAAP. By using a radiative transfer model, the light absorption coefficients can be calculated. The instrument design offers the advantage of accounting for

the multiple scattering effects and is able to measure absorption coefficients at three different



wavelengths, as initially introduced, and was later upgraded to measure at five different wavelengths (375, 407, 532, 635, and 850 nm) (Massabò et al., 2015). The MAAP aerosol-laden filter tape was collected at the ATTO site and analyzed by MWAA at the University of Genoa, Genoa, Italy. During transport of the samples, they could be affected by aging of the organic aerosol, microbial processes

and/or loss of semi-volatile material (Laskin et al., 2015; Saleh et al., 2014). In order to avoid these issues, the samples were collected everyday directly from the MAAP and kept frozen (-4 °C) during the campaign time and transported in a cool bag with blue ice (~72 h) to the laboratory in Genoa. We reanalyzed some samples after being stored at room temperature during three days to investigate the potential aging of carbonaceous material collected on the filters and found no significant differences in

the absorbance results measured by the MWAA.

**2.3 Aethalometer measurements and corrections**

The Aethalometer continuously measures light attenuation on an aerosol-laden filter. The attenuation is defined, according to the Lambert-Beer law, as

$$ATN = 100\% \cdot \ln\left(\frac{I}{I_0}\right) \qquad (2)$$

where $I$ and $I_0$ are the light intensity transmitted through an aerosol loaded and an original area of the filter tape, respectively. A list of symbols and acronyms is provided in Table 1. The instrument is programmed to calculate the equivalent black carbon ($BC_e$) mass concentration by assuming that a change in attenuation ($\Delta ATN$) is caused by an increase in the BC mass deposited on the filter substrate during an interval $\Delta t$ (min), as follows:

$$BC\,(ng/m^3) = \frac{A \cdot \Delta ATN}{\alpha_{ATN} \cdot Q \cdot \Delta t} \qquad (3)$$

where $A$ is the filter area (1.67 cm²), $\alpha_{ATN}$ is the BC mass attenuation cross-section in m² g⁻¹ (14625/λ) and $Q$ is the volumetric flow rate in l min⁻¹. By using the $\alpha_{ATN}$ recommended by the manufacturer, we reversed the calculation to convert reported mass concentrations back to attenuation coefficients, as

$$\sigma_{ATN} = BC\,(ng/m^3) \cdot \alpha_{ATN} \qquad (4)$$

15                                            8



Two different correction schemes were applied to our dataset, including the Schmid et al. (2006) and the

Collaud Coen et al. (2010) algorithms. These two correction schemes were chosen because both of them

compensate for the three artifacts that affect Aethalometer measurements. The Arnott and Collaud

Coen's Arnott-like corrections were excluded due to their limitations when dealing with low-absorption

data. Both, Collaud Coen and Schmid corrections, require concomitant scattering measurements and a

reference absorption measurement, which in our case was the MAAP. Moreover, we present and discuss

a comparison of corrected Aethalometer data to the multi-wavelength light absorption measurement

obtained from the MWAA.

### 2.3.1 Schmid correction algorithm

The Schmid correction consists of an iterative procedure, which is applied to each measured attenuation

spectrum, as explained by Rizzo et al. (2010). As a first step, the reference $C$ ($C_{ref}$) is calculated for

attenuation coefficients corresponding to attenuation values lower than 10 %, when the filter-loading

correction is considered negligible (ATN < 10 %; R ≈ 1). By using MAAP absorption coefficient

measurements, it is possible to obtain $C_{ref}$ as follows:

$$C_{ref} = \frac{\sigma_{ATN,10}}{\sigma_{MAAP}} \qquad (5)$$

where $\sigma_{MAAP}$ is the absorption coefficient measured by the MAAP at 637 nm and $\sigma_{ATN, 10}$ is the

attenuation coefficient at 637 nm ($\lambda_{MAAP}$) when ATN < 10 %.

Attenuation coefficients at 590 nm were interpolated to 637 nm as,

$$\sigma_{ATN}(\lambda_{MAAP}) = \sigma_{ATN}(590\,nm) \cdot \left(\frac{\lambda_{MAAP}}{590\,nm}\right)^{-\mathring{a}_{ATN}} \qquad (6)$$

The attenuation Ångström exponent $\mathring{a}_{ATN}$ used in this step was calculated by applying a log-log fit to

$\sigma_{ATN}$ vs. λ, where $\mathring{a}_{ATN}$ was obtained from the slope as follows:

$$\ln \sigma_{ATN} = -\mathring{a}_{ATN} \ln(\lambda) + \ln(constant) \qquad (7)$$

Absorption Ångström exponents ($\mathring{a}_{ABS}$) were obtained in a similar way in further calculations.

The multiple scattering correction factor, $C_{ref}$, obtained from Eq. (5) was averaged to calculate the

measured filter-loading correction factor, $R_{meas}$, as




$$R_{meas} = \frac{\sigma_{ATN}}{\sigma_{MAAP} \cdot \overline{C}_{ref}}$$

(8)

Weingartner et al. (2003) found that the linear relationship between $\sigma_{ATN}$ and ln(ATN) can be used to parameterize the filter-loading effect. The slope of this relationship was given by a parameter called the *shadowing factor, f*. By applying a linear fit to Eq. (8) vs. ATN data, it is possible to obtain the shadowing factor as follows


$$R = \left(\frac{1}{f} - 1\right)\left(\frac{\ln ATN - \ln 10}{\ln 50 - \ln 10}\right) + 1$$

(9)

Assuming $f$ is wavelength independent, the averaged $f$ is used to calculate $R$ from Eq. (9) at different wavelengths.

The next step is the parameterization of $C$ as a function of the AAE. In order to do that, $C$ was rewritten as:


$$C = C^* + m_s \frac{\omega_0}{1 - \omega_0}$$

(10)

where $C^*$ corresponds to the multiple scattering effect by filter fibers, $m_s$ to the aerosol scattering effect and $\omega_0$ to the single scattering albedo. The $C^*$ and $m_s$ values were taken from Arnott et al. (2005) (see Table S1), and the $\omega_0$ was calculated at 637 nm using measured scattering and absorption coefficients at 637 nm. The values of $\omega_0$ at different AE wavelengths were obtained by using the following equation:


$$\omega_0(\lambda) = \frac{\omega_{0,ref}\left(\frac{\lambda}{\lambda_{ref}}\right)^{-\mathring{a}_{SCA}}}{\omega_{0,ref}\left(\frac{\lambda}{\lambda_{ref}}\right)^{-\mathring{a}_{SCA}} + \left(1 - \omega_{0,ref}\right)\left(\frac{\lambda}{\lambda_{ref}}\right)^{-\mathring{a}_{ABS}}}$$

(11)

Using different AAE ($\mathring{a}_{ABS} = 1; 1.25; 1.5; 1.75; 2$), $C$ was calculated and the obtained values were used to plot ln($C$) vs. ln($\lambda$). The coefficients resulting from a quadratic fit were used to parameterize $C$ as a function of AAE (see Fig. 4 in Schmid et al. (2006)). The AAE resulting from the iterative procedure converged after seven iterations in our calculations.

## 2.3.2 Collaud Coen correction algorithm

In this study we implemented the Collaud Coen correction algorithm that resembles the Schmid correction (see eq. 14b in Collaud Coen et al. (2010)). This algorithm is different from the original





Schmid algorithm in the calculations of the filter-loading effect and the multiple scattering correction factor.

Regarding the filter-loading effect, Collaud Coen et al. used the linear dependency of the shadowing factor, $f$, on the single scattering albedo, $\omega_0$, expressed by Eq. (12), to calculate $f$ using measured $\omega_0$ and assuming $m$ was constant ($m = 0.74$).

$$f = m \cdot (1 - \omega_0) + 1 \qquad (12)$$

Additionally, they found statistically better results by correlating $\sigma_{ATN}$ vs. ATN, instead of the

logarithmic correlation proposed by Weingartner et al. (2003), which was implemented by Schmid et al. (2010). Considering no filter-loading artifact for ATN = 0, they proposed the following equation, which replaces Eq. (9):

$$R = \left( \frac{1}{m(1 - \bar{\omega}_0) + 1} - 1 \right) \left( \frac{ATN}{50} \right) + 1 \qquad (13)$$

In this case, $\omega_0$ was averaged for every filter spot period (from one filter spot change to the next) and

this average was used for calculating every measurement included in the corresponding filter spot period. The different $\omega_0$ values at different wavelengths were calculated by using Eq. (11) but including attenuation Ångström exponents ($\mathring{a}_{ATN}$) because $\mathring{a}_{ABS}$ is not known yet.

Filter-loading corrected data is then divided by the MAAP absorption coefficients to obtain an average $C_{ref}$. Regarding the embedded aerosol scattering effects, the Collaud Coen correction includes a change

in the aerosol scattering effect parameter expressed as $m_s$ in Eq. (10). The constant $m_s$ values used by Schmid et al. (2010) correspond to ammonium sulfate. Collaud Coen substituted them by the measured aerosol scattering properties by using the following equation

$$m_s = \beta_{SCA}^{(d-1)} \cdot c \cdot \lambda^{(-\mathring{a}_{SCA}(d-1))}; \qquad (14)$$

$d = 0.564;$

$c = 0.79732$ ($\sigma_{sp}$ in Mm$^{-1}$ units)

where $\beta_{SCA}$ is the scattering proportionality constant and $c$ and $d$ are constants corresponding to the power-law relation between $\sigma_{ATN}$ and $\sigma_{sp}$, previously reported by Arnott et al. (2005).





### 2.3.3 Considerations for the algorithms' comparison

Both algorithms used in this study calculate $C_{ref}$ in different ways. As shown in Eq. (5), the Schmid
algorithm filters the data for ATN < 10 % in order to account only for the scattering by filter fibers in
the $C_{ref}$ calculation. On the other hand, Collaud Coen algorithm applies a prior filter-loading correction
and then, by dividing the reference absorption data (MAAP) by the Aethalometer attenuation
coefficients, they obtain $C_{ref}$, which accounts for both, scattering by filter fibers and scattering by
embedded aerosol particles. The latter approach makes it difficult to separate the contributions by filter
fibers and aerosol particles to the multiple scattering compensation. Therefore, we decided to make use
of the Schmid formulation when analyzing and presenting $C_{ref}$ in this work; i.e., $C_{ref}$ only accounting for
scattering by filter fibers.

### 3 Results and discussion

The beginning of the sampling period is characterized by low scattering coefficients compared to the
second half of the period when scattering increases significantly. Several scattering peaks can be
observed after the beginning of August (see Fig. 1a). Occasionally, local or regional biomass burning
plumes reach the site during the dry season and scattering by aerosol particles increases significantly
due to enhanced concentration of fine mode aerosol particles, which are more efficient in scattering
light in the visible range. The major effect of multiple scattering artifacts is evident when comparing
MAAP measured absorption coefficients and Aethalometer measured attenuation coefficients (see Fig.
1b). The absorption coefficients averaged $1.8 \pm 2.1$ $Mm^{-1}$, with the minimum values occurring in the
beginning of the sampling period, whereas a maximum of absorption (up to 15.9 $Mm^{-1}$, measured by
MAAP) took place between 18 and 23 August 2014. Calculated back-trajectories using the HYSPLIT
model (Draxler and Hess, 1998) confirmed that air masses on the days of maximum absorption and
scattering were coming from south and south east, an area with intense fire activity, see supplementary
material, Fig. S1. Levoglucosan measurements further confirmed the predominance of biomass burning
originated aerosol particles (not shown). From 01 June to 01 August 2014, the attenuation coefficient at
637 nm had a median of 5.1 $Mm^{-1}$ (3.2 – 7.9, interquartile range, IQR). Then, during the first days of



August, it increased slightly until the biomass burning event took place on 18 – 23 August 2014. The

maximum attenuation coefficient during this event reached 115 Mm$^{-1}$. Details about this event, regarding chemical composition and CCN activity, are presented in Pöhlker et al. (2016a, 2016b). The observed absorption and attenuation coefficients represent typical conditions at the ATTO site for the wet, transition and dry periods. In the next sections, we present data compensated to account for the different filter artifacts and study the influence of the applied compensation algorithms on the AAE. The

artifacts that affect the AAE retrieval from filter-based multi-wavelength absorption measurements could be avoided by using photoacoustic spectrometric (PAS) methods that have been successfully implemented to measure light absorption by suspended aerosol particles (e.g., Ajtai et al., 2010). However, PAS measurements have high detection limits and have only been implemented at near-source measurement sites (Cappa et al., 2012; Cheng et al., 2016; Lewis et al., 2008) and not in clean

environments like central Amazonia.

### 3.1 Aethalometer corrections

Immediately after every Aethalometer filter change, aerosol particles are collected on a clean new spot. Under these conditions, the filter-loading effect is considered to be negligible because there is not enough aerosol on the filter to "darken" the substrate (Virkkula et al., 2007). Therefore, the only bias to

the Aethalometer response is given by the scattering effects by filter fibers. The scattering by filter fibers, expressed as $C_{ref}$, was calculated by using Eq. (5), assuming $R \approx 1$ for data corresponding to ATN < 10 %. The $C_{ref}$ time series is shown in Fig. 2. We observed that $C_{ref}$ decreased somewhat from June – July to August – September when the average ± one standard deviation values were 6.3 ± 1.5 and 4.9 ± 1.1, respectively. Additionally, we observed a larger $C_{ref}$ variability during the transition period, which

may increase the uncertainty of the corrected absorption coefficients. This seasonal effect on the multiple scattering compensation parameter could be related to the condensation or adsorption of semi-volatile organic compounds or liquid organic aerosol particles on the filter fibers, inducing a change in the filter matrix optical properties (Collaud Coen et al., 2010; Subramanian et al., 2007; Weingartner et al., 2003). The Schmid algorithm uses an average $C_{ref}$ for further calculation of the filter-loading

correction factor, $R$. We found that using an overall average $C_{ref}$ significantly affects the calculation of





the shadowing factor ($f$). Therefore, two different averages of $C_{ref}$ were implemented in this work for the two above-mentioned periods, transition (June – July) and dry season (August – September). Subsequent multiple scattering correction calculations were conducted using real-time $C_{ref}$ values. The measured filter-loading calibration factor ($R_{meas}$) was obtained by using Eq. (8). Then, by following

the Schmid algorithm, the shadowing factor was calculated by applying a fit to equations (8) and (9) (Rizzo et al., 2011; Schmid et al., 2006). The calculated average shadowing factors were $1.10 \pm 0.10$ and $1.04 \pm 0.08$ for June-July and August-September, respectively. These values were lower compared to those obtained for darker aerosols ($f = 1.23 – 1.89$) (Weingartner et al., 2003) and for biomass burning aerosol ($f = 1.2$) (Schmid et al., 2006). The filter-loading correction calculation resulted in R

correction factors of $0.98 \pm 0.02$ and $0.99 \pm 0.01$ at 880 nm, for June – July and August – September, respectively. A slight wavelength dependence was observed; the $R$ values were up to 4% higher at 370 nm compared to those calculated at 880 nm during the cleanest period of this study (June – July). A similar behavior was observed during August – September. As explained by Schmid et al. (2006), this wavelength dependency is related to the fact that $R$ depends on ATN, which increases with decreasing

wavelength. The obtained $R$ correction factors were very close to 1, i.e., the filter loading effect barely affected the conversion from attenuation to absorption coefficients, even during the most polluted period, August – September. A filter-loading correction factor close to 1 was expected since the average $\omega_0$ measured during the campaign was $0.88 \pm 0.04$ at 637 nm. A high $\omega_0$ is related to the predominance of scattering aerosol particles, which diminishes the shadowing effect of dark aerosol particles

embedded in the filter matrix (Weingartner et al., 2003).

To compare both correction schemes in terms of the filter-loading correction, $C_{ref}$ was recalculated after compensating all the data for filter loading by: 1) following the Schmid et al. correction and, 2) the Collaud Coen et al. correction, which includes $\omega_0$ in the shadowing factor calculation and the relationship $\sigma_{ATN}$ vs. ATN. We found no statistical difference between the two correction algorithms in

terms of the filter-loading compensation because this effect was generally low over the sampling period. More information about the effect of increasing attenuation on the calculated $C_{ref}$ after applying the filter-loading correction can be found in the supplementary material (Figure S2).





As already mentioned, the multiple scattering effects significantly affect the correction of Aethalometer

data by a factor of 5 to 7. According to previous studies, the multiple scattering correction is the most

important one in ambient aerosol with a high $\omega_0$ (Collaud Coen et al., 2010; Rizzo et al., 2011; Schmid

et al., 2006; Segura et al., 2014). The seasonal variability of $C$ can be explained by the different

scattering properties of the aerosol particles in the different seasons (Collaud Coen et al., 2010). In order

to compare the different scattering contributions to $C$, we calculated $C_{ref}$ and $C_{sca}$ by using the Collaud

Coen algorithm. $C_{sca}$ was calculated using Eq. (14) in this case. We observed that a lower $\omega_0$ during the

biomass burning period was related to a decrease in the scattering correction factor, $C_{sca}$. Given that $C_{ref}$

decreased simultaneously with $C_{sca}$, the relative contribution of each, $C_{sca}$ and $C_{ref}$, was examined. The

relative contribution from the scattering correction decreases with decreasing $\omega_0$. In contrast, it

decreases with increasing $\beta_{sca}$, see Fig. 3. No correlation was found between $C_{sca}$ and $\mathring{a}_{sca}$ since the

scattering Ångström exponent was quite stable during the sampling period with the exception of the few

days influenced by regional biomass burning (see Fig. S3). In other words, the $C_{sca}$ relative contribution

was only affected by variations on $\beta_{sca}$ and $\omega_0$. Given that R is almost negligible in our dataset, the

comparison between both algorithms was done in terms of their different ways to treat the multiple

scattering effects.

A scatter plot of both corrections' outputs vs. MAAP measurements is shown in Fig. 4. We found that

corrected AE data fitted very well the MAAP measurements in the case of the Schmid correction, with a

slope of 1.04 (1.02 – 1.05); i.e, the Schmid correction overestimates the absorption coefficient by only 2

– 5 %. In the case of the Collaud Coen correction, it was found that the AE corrected absorption

coefficients were underestimated by 19 – 21 %. According to our calculations, the Collaud Coen

algorithm tends to produce higher $C_{ref}$ values than the Schmid algorithm. The difference between both

correction schemes in terms of the comparison to MAAP measurements can be related to the

parameterization of $C$ applied by Schmid et al., which is not implemented by Collaud Coen et al., and

the way Collaud Coen et al. estimate $C_{ref}$.



### 3.2 Absorption Ångström Exponent

The MWAA was used as a reference multi-wavelength measurement since it accounts for multiple
scattering effects by means of a similar configuration to the MAAP. Light absorption coefficients
obtained from the MWAA (at 635 nm) and from the MAAP (at 637 nm) were compared by applying an
linear regression to both datasets after integrating the MAAP data over the filter total sampling times.
The fit resulted in an MWAA underestimation by 14 to 18% when fitting the whole dataset. However,
when comparing only data from the polluted period (18 – 23 August 2014), the MWAA underestimation
was only ~5 %. A scatter plot, including the fits, can be seen in Fig. 5. The MWAA underestimation for
low absorption coefficient samples might be related to the proximity to the instrument detection limits.
The possibility of losing part of the BrC aerosol of medium volatility was also considered and all data
with $\sigma_{ap} < 1$ Mm$^{-1}$ were considered with caution when making any interpretation in the further analysis.
MWAA data measured at five different wavelengths was used to retrieve $å_{abs}$ by applying a log-log fit as
expressed in Eq. 7. Figure 6 shows the MWAA Ångström exponents and their uncertainty intervals,
together with the values obtained from the two different Aethalometer corrections. The MWAA $å_{abs}$
retrieved from each filter were not all statistically optimal; 30 out of 175 had a R² < 0.85. All the values
below this R² limit were excluded from the results shown in Fig. 6. Absorption Ångström exponents
obtained using the Schmid correction were mostly higher than the MWAA results. On the other hand,
the Collaud Coen correction resulted in a better approach to reproduce the MWAA data, with most of
the results in the MWAA uncertainty range. The original attenuation Ångström exponent was also found
to fit very well the MWAA-retrieved AAE, see Fig. S4. During the biomass burning period, from 18 to
23 August 2014, the BrC contribution became more important and caused an increase in the AAE and
both algorithms' results became similar to each other and slightly higher than the MWAA $å_{abs}$. After the
biomass burning episode, when the scattering and absorption coefficients fell down to background
levels, the offset between both algorithms, in terms of $å_{abs}$, widened again. In this regard, the Collaud
Coen algorithm, which includes a modified scattering correction, seems to be more appropriate to
retrieve the AAE for a broader range of  absorption coefficients.
A scatter plot of the AAE data, including the corresponding linear fits, is shown in Fig. 7. The data
analyzed in this comparison includes only filters that had a $\sigma_{ap}$ vs. λ log-log fit with R² > 0.85. Although





both algorithms overestimate the AAE retrieved from the MWAA measurements, the Collaud Coen algorithm produces a lower offset and a better linear fit, with a $R^2 = 0.72$. On the other hand, the Schmid algorithm seems to be artificially enhancing the absorption at lower wavelengths. When applying linear regressions forced through the origin, the overall tendency showed a statistically

significant AAE overestimation by the Schmid algorithm and a better fit for the Collaud Coen algorithm (not shown).

### 3.3 Overestimation of near-UV absorption by AE corrections

The unexpectedly high AAE, especially that obtained by applying the Schmid algorithm, is probably caused by an artificial enhancement of the near-UV absorption. Figure 8 shows the relative

enhancement of the absorption coefficients at 370 nm, compared to the MWAA absorption at 375 nm. No interpolation was applied to match both wavelengths since they are close enough that the differences are negligible (~3 % for an AAE of 2.0). It is clear that the Schmid algorithm almost always overestimated the absorption at 370 nm. Only a few filters showed a difference close to or below zero. On average, the Schmid algorithm overestimation relative to MWAA was a factor of $0.46 \pm 0.31$. In the

case of the Collaud Coen algorithm, the average difference was slightly negative (underestimation), being a factor of $-0.07 \pm 0.30$, and reaching even lower values during the biomass burning event. A near-UV over- or underestimation of the data, will substantially affect brown carbon calculations, if apportionment algorithms based on the wavelength dependence of absorption are used. A BrC estimation is beyond the scope of this paper. However, we suggest critically evaluating corrected

Aethalometer data when using it to retrieve BrC / BC contributions.

### 3.4 Adaptation of the Collaud Coen algorithm

As shown before, the Collaud Coen algorithm underestimates absorption coefficients compared to the reference absorption measurement (MAAP). The reason for this underestimation might be related to the way the algorithm calculates $C_{ref}$ by dividing Aethalometer filter-loading corrected data by MAAP data.

By doing this, the calculated $C_{ref}$ accounts for the scattering by embedded aerosol particles and filter fibers when it should only account for the scattering by filter fibers as explained in section 2.3.3.



Therefore, we considered to test the Collaud Coen algorithm by calculating $C_{ref}$ as in the original Schmid correction, using Eq. (5), but calculating it only for ATN values < 5 % to make sure only the multiple scattering by filter fibers was responsible for the measurement artifact. The result of this

modified Collaud Coen correction scheme is illustrated by Fig. 9. We found that using this approach the Collaud Coen corrected data underestimates the MAAP data by only 6 – 9 %, which is within the standard deviation of the measurement. The correlation coefficient ($R^2 = 0.88$) was similar to that obtained for the comparison Schmid – MAAP (Fig. 4, $R^2 = 0.89$), however. The adapted Collaud Coen algorithm did not affect the retrieved AAE values and it was significantly correlated with the MWAA-

retrieved AAE (slope: 1.14 (1.03 – 1.26) and $R^2 = 0.72$), similar to the values obtained for the unmodified Collaud Coen algorithm (see Fig. 7). Although the modified Collaud Coen algorithm does not provide a better approach to retrieve AAE from Aethalometer data compared to the original algorithm, it improves the comparison to MAAP measurements at 637 nm.

**4 Conclusions**

We applied two different correction algorithms to compensate for the various Aethalometer absorption measurement artifacts. The compensated data was compared to an offline multi-wavelength reference absorption measurement technique. This comparison allowed studying the effects of the correction schemes on the absorption at lower wavelengths and showed how this affects the AAE retrieval. We found that the Schmid algorithm is a good approach to reproduce the reference MAAP absorption

coefficients from Aethalometer data but overestimates the AAE compared to that obtained by the multiple wavelength measurement (MWAA). On the other hand, the Collaud Coen algorithm as well as the "raw" Aethalometer attenuation spectral dependence reproduced quite well the AAE values obtained from MWAA measurements. The data analysis indicates that the "live" scattering correction implemented in the Collaud Coen algorithm is appropriate to achieve more realistic absorption

coefficients at lower wavelengths. However, the Collaud Coen algorithm underestimated absorption at 637 nm when compared to MAAP. This offset was reduced by slightly modifying the algorithm in the calculation of the multiple scattering correction factor. The adapted algorithm was found more appropriate to achieve the best correlations with the MAAP absorption coefficients. The under- or



overestimation of short-wavelength absorption coefficients by compensation algorithms is a factor that

has to be considered when using corrected Aethalometer data to apportion the black and brown carbon

contributions to total absorption.

## 5 Acknowledgments

This work has been supported by the Max Planck Society (MPG) and the Max Planck Graduate School (MPGS). For the operation of the ATTO site, we acknowledge the support by the German Federal

Ministry of Education and Research (BMBF contract 01LB1001A) and the Brazilian Ministério da Ciência, Tecnologia e Inovação (MCTI/FINEP contract 01.11.01248.00) as well as the Amazon State University (UEA), FAPEAM, LBA/INPA and SDS/CEUC/RDS-Uatumã. P. A. acknowledges support from FAPESP – Fundação de Amparo à Pesquisa do Estado de São Paulo. J. S. is grateful for a PhD scholarship from the Fundación Gran Mariscal de Ayacucho (Fundayacucho). We acknowledge Paola

Fermo, Raquel Gonzalez and Lorenza Corbella for the levoglucosan analysis. This paper contains results of research conducted under the Technical/Scientific Cooperation Agreement between the National Institute for Amazonian Research, the State University of Amazonas, and the Max-Planck-Gesellschaft e.V.; the opinions expressed are the entire responsibility of the authors and not of the participating institutions. We highly acknowledge the support by the Instituto Nacional de Pesquisas da

Amazônia (INPA). We would like to especially thank all the people involved in the technical, logistical, and scientific support of the ATTO project, in particular Reiner Ditz, Jürgen Kesselmeier, Niro Higuchi, Matthias Sörgel, Stefan Wolff, Thomas Disper, Andrew Crozier, Uwe Schulz, Steffen Schmidt, Antonio Ocimar Manzi, Alcides Camargo Ribeiro, Hermes Braga Xavier, Elton Mendes da Silva, Nagib Alberto de Castro Souza, Adi Vasconcelos Brandão, Amaury Rodrigues Pereira, Antonio Huxley Melo

Nascimento, Thiago de Lima Xavier, Josué Ferreira de Souza, Roberta Pereira de Souza, Bruno Takeshi, Ana María Yáñez-Serrano and Wallace Rabelo Costa. Moreover, we thank Thorsten Hoffmann, Ulrich Pöschl, Arthur Sedlacek, Jeannine Ditas, Su Hang, Jian Wang, Sachin Gunthe, Jan-David Förster, Ming Jing, Tobias Könemann, Maria Praß, Andrea Arangio and Bruna Amorim Holanda for support and stimulating discussions.



The authors gratefully acknowledge the NOAA Air Resources Laboratory (ARL) for the provision of
the HYSPLIT transport and dispersion model and READY website (http://www.ready.noaa.gov) used in
this publication.

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

50



**Table 1.** List of used symbols and acronyms

| Description | Acronym | Symbol | Units |
|---|---|---|---|
| Attenuation | ATN | ATN | |
| Absorption Ångström exponent | AAE | $\mathring{a}_{ABS}$ | |
| Scattering Ångström exponent | SAE | $\mathring{a}_{SCA}$ | |
| Attenuation Ångström exponent | | $\mathring{a}_{ATN}$ | |
| Attenuation coefficient | | $\sigma_{ATN}$ | $\mathrm{m}^{-1}$ |
| Absorption coefficient | | $\sigma_{ap}$ | $\mathrm{m}^{-1}$ |
| Scattering coefficient | | $\sigma_{sp}$ | $\mathrm{m}^{-1}$ |
| Mass attenuation cross-section | | $\alpha_{ATN}$ | $\mathrm{m}^2\,\mathrm{g}^{-1}$ |
| Mass absorption cross-section | MAC | $\alpha_{ABS}$ | $\mathrm{m}^2\,\mathrm{g}^{-1}$ |
| Scattering proportionality constant | | $\beta_{SCA}$ | $\mathrm{m}^{-1}$ |
| Filter-loading correction factor | | $R$ | |
| Shadowing factor | | $f$ | |
| Multiple scattering correction factor | | $C_{ref}$ | |
| Scattering correction factor | | $C_{sca}$ | |
| Scattering effect parameter | | $m_s$ | |





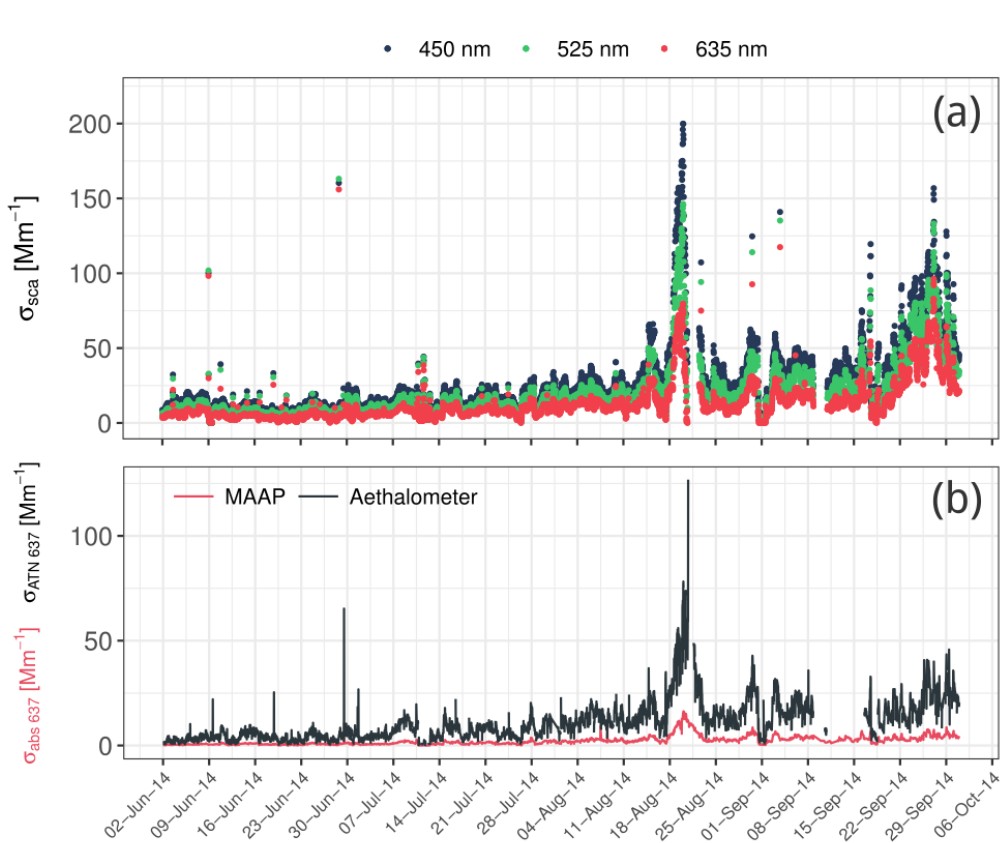

**Figure 1.** Time series (June – September 2014) of a) scattering by aerosol particles measured by the nephelometer and b) Aethalometer attenuation and MAAP absorption coefficient measurements at 637 nm during the sampling period.



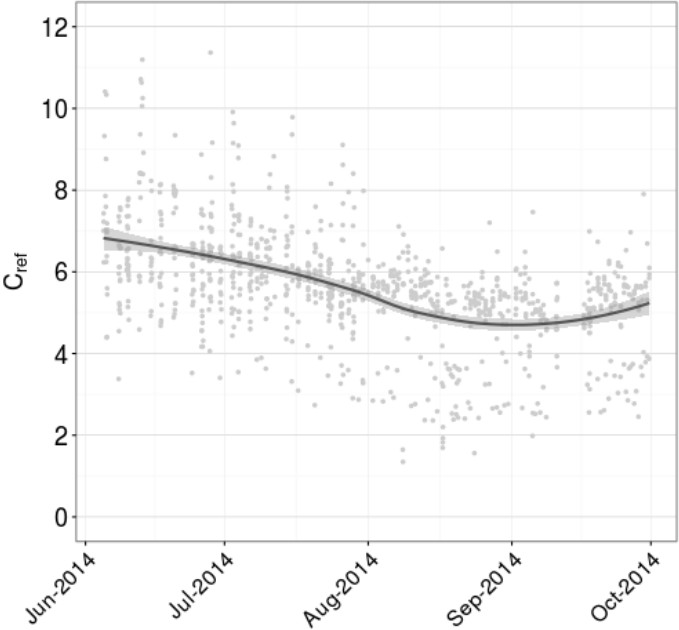

**Figure 2.** Multiple scattering correction calculated by using MAAP absorption coefficients as reference ($\lambda$ = 637 nm). Light gray points represent all calculated $C_{ref}$ values. The black line and shaded area represent a conditional non-parametric mean estimation and its confidence limits, respectively.



**Figure 3.** Filter cycle averaged data corresponding to a) scattering proportionality constant, b) single scattering albedo at 660 nm, and c) relative contribution of $C_{ref}$ and $C_{sca}$ to the total multiple scattering compensation ($C_{ref} + C_{sca}$) at 660 nm. Vertical bars in a) and b) correspond to one standard deviation.



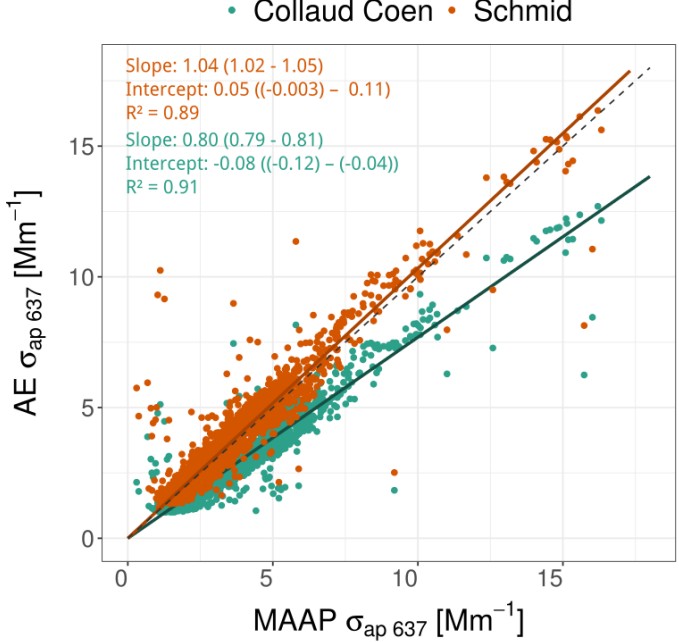

**Figure 4**. Scatter plot of different Aethalometer corrections results vs. MAAP absorption coefficients (all data at 637 nm). The fit was obtained by applying a standardized major axis regression. The fit slopes include the limits to the 95% confidence intervals in brackets. The dashed grey line represents a 1:1 relationship.





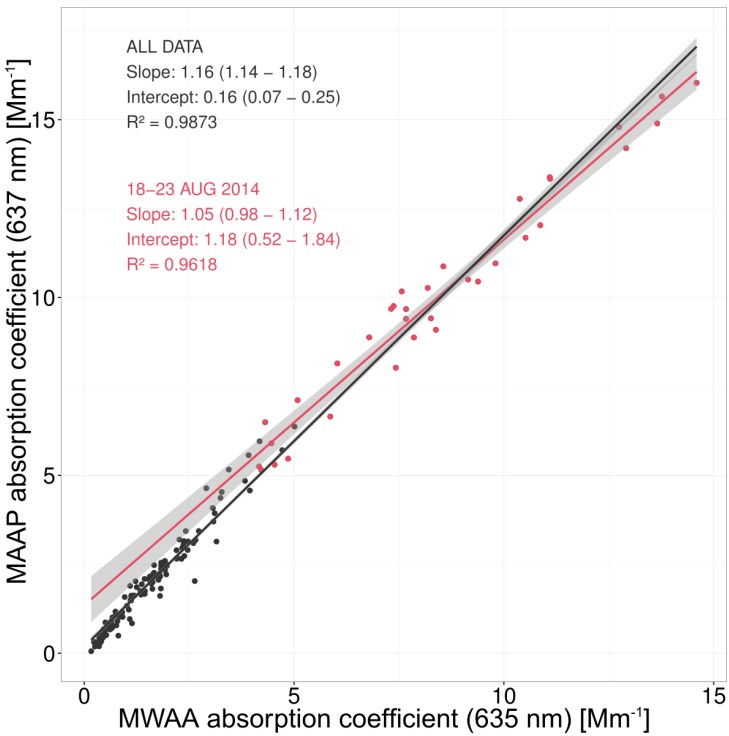

**Figure 5.** Linear regression applied to MWAA and MAAP absorption coefficients at 635 and 637 nm, respectively. For a) the whole campaign period and b) 18 – 23 August 2014. The 95 % confidence intervals of the fits are shown as shaded areas. The fit was obtained by applying a standardized major axis regression.



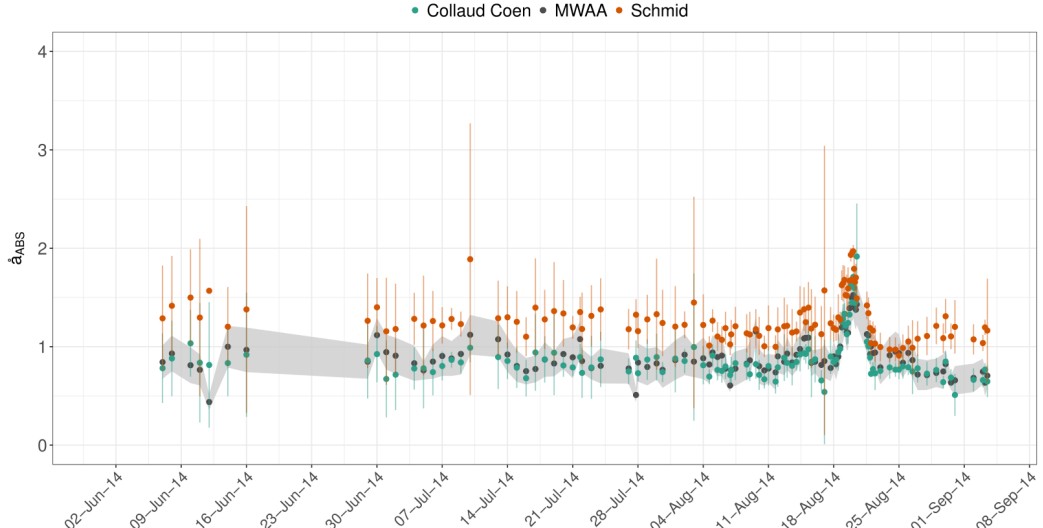

**Figure 6.** Absorption Ångström exponent retrieved from Schmid and Collaud Coen correction algorithms averaged over MWAA sample intervals. Vertical lines on the different correction points correspond to one standard deviation. Black circles represent MWAA obtained $\mathring{a}_{abs}$. The gray shadow corresponds to the standard error of the logarithmic fit.





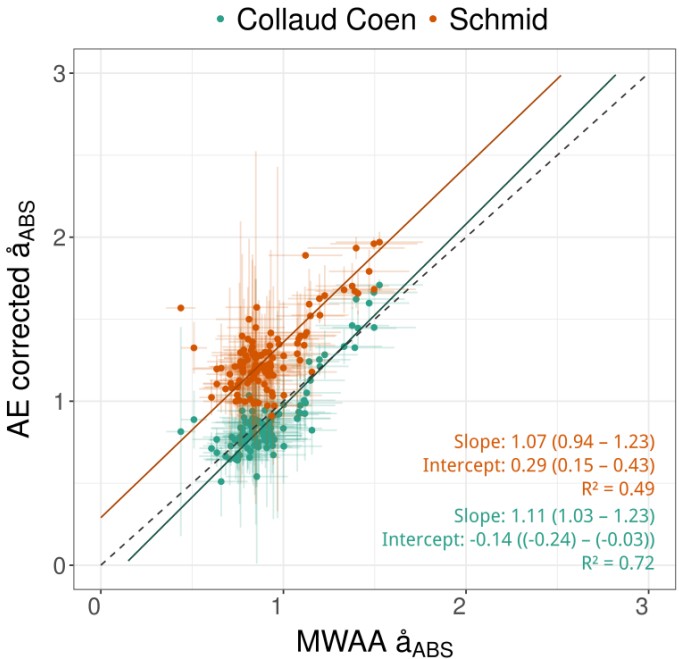

**Figure 7.** Scatter plot of AAE values obtained by Aethalometer corrections vs. AAE obtained from MWAA measurements. The dark-coloured lines correspond to the standardized major axis linear fits and light-coloured lines correspond to one standard deviation of the retrieved AAE data. The dashed gray line represents a 1:1 relationship.





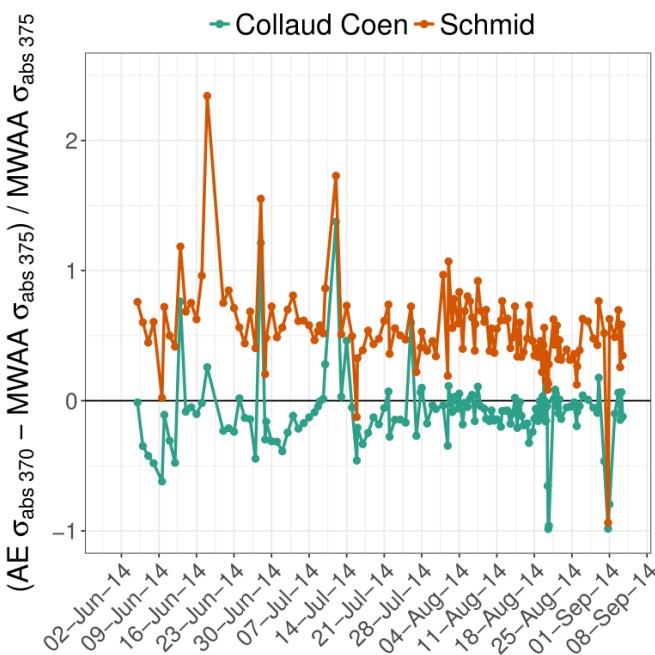

**Figure 8.** Overestimation of Aethalometer corrected absorption coefficients relative to MWAA at 370 nm. Values above zero are related to an overestimation of $\sigma_{ap}$ and, below zero, to an underestimation of $\sigma_{ap}$ at this given wavelength.





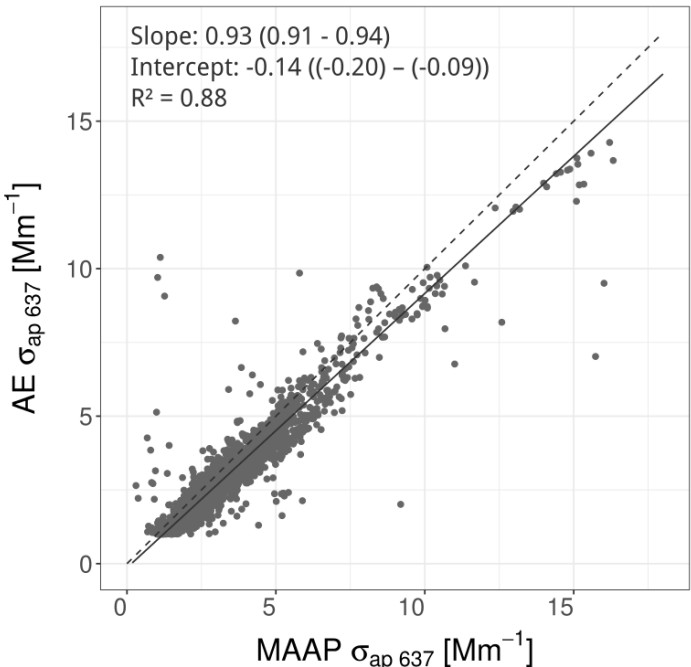

**Figure 9.** Scatter plot of Aethalometer corrected data by using the modified Collaud Coen algorithm vs. MAAP absorption data. The black continuous line represents the result of applying a linear fit, using a standardized major axis estimation. The dashed line represents a 1:1 relationship.