# Peer review of "Comparison of different Aethalometer correction schemes and a reference multi-wavelength absorption technique for ambient aerosol data"

_Atmospheric Measurement Techniques, 2016_

## Referee Comment (RC1) · Anonymous Referee #1 · 8 Jan 2017

The paper presents an analysis of multiwavelength absorption data collected at an Amazonian site. Aerosol optical properties were measured with an aethalometer, MAAP and a nephelometer. The MAAP filter spots were later analyzed with an offline method, the MWAA that was considered here as the absorption standard. The MWAA is based on the same principle as the MAAP since it measures both transmitted and scattered light and a radiative transfer algorithm similar to that in the MAAP is applied to calculate the absorption coefficient. The good point is that it has several wavelengths, the weak points are that it is still a filter-based method with related artifacts and that its time resolution is not as good as that of online instruments.

The aethalometer data were processed with two methods, the Schmid et al. (2006) and

the Collaud Coen et al. (2010) algorithm and the important Cref value was retrieved for both methods. The analysis shows that there are large sources of uncertainty of Cref and the wavelength dependency of absorption (AAE). One of the algorithms seems to be better in one respect, the other in another way. Absolute truth is still not found.

My first question is related to Eq. (11). On line 256 you write that you use five different AAEs to calculate SSA and further Cref. Does this not result in five different Crefs? Do you give the average as the final Cref? Why not using the calculated AAEs instead of five fixed values? The whole procedure is not clear enough and unambiguously explained so that I would try to apply it to my own data.

Line 256. "Using different AAE ($å_{abs}$ = 1,..." Why do you use the symbols AAE and $å_{abs}$ for the same thing? Be consistent throughout the text.

L384-389 "A scatter plot of both corrections' outputs vs. MAAP measurements is shown in Fig. 4. We found that corrected AE data fitted very well the MAAP measurements in the case of the Schmid correction, with a slope of 1.04 (1.02 – 1.05); i.e, the Schmid correction overestimates the absorption coefficient by only 2– 5 %. In the case of the Collaud Coen correction, it was found that the AE corrected absorption coefficients were underestimated by 19 – 21 %." I don't understand. My first thought is that the Crefs are just wrong. As far as I have understood the whole thing of getting Cref is based on forcing the aethalometer-derived absorption to match the MAAP-derived absorption. Should not the slope should be very close to if not exactly 1 regardless of the algorithm selected? Please explain.

L439-440. It is reminded to evaluate critically the corrected filter-based absorption data when using it to retrieve BrC / BC contributions. There are quite a few papers that discuss this same issue. Give some references to such papers, some more credit to earlier work could be given.

Fig 3. Beta should have units. scattering coefficient = beta x lambdaˆ(-alpha)

[Figure]

Figures 6 and S4 are just the same with the exception that in S4 there is one more line. Why would you not show it simply in Fig 6 and omit S4?

---

## Referee Comment (RC2) · Anonymous Referee #2 · 18 Jan 2017

Review of amt-2016-361 on "Comparison of different Aethalometer correction schemes and a reference multi-wavelength absorption technique for ambient aerosol data" by Jorge Saturno et al.

GENERAL COMMENT

The study targets the measurement of absorption Ångström exponents (AAE) by filter-based multi-wavelength light absorption measurement methods. Light attenuation data from the widely used 7-wavelength Aethalometer are analyzed by applying two correction schemes according to Schmid et al. (2006) and Collaud Coen et al. (2010) and comparing the results to data from the filter-based off-line Multi-Wavelength Absorbance Analyzer MWAA (Massabò et al., 2013; Massabò et al., 2015). Reference methods of the measurement of light scattering and light absorption were a 3-$\lambda$ Integrating Nephelometer (Model Aurora 3000, Ecotech) and a single-wavelength Multi-Angle Absorption Photometer MAAP (Model 5012, Thermo Electron Group). Data have been collected at the ATTO site in Amazonia from the wet-to-dry transition season to the dry season in 2014.

The topic of the study is of relevance for the research area of determining black carbon (BC) and brown carbon (BrC) from the wavelength dependence of the light absorption coefficient, characterized by the AAE. The presented data is well suited for the study and of high importance and thus deserves publication in AMT. However, before being suitable for publication, the manuscript requires major revisions which are highlighted in the following.

SPECIFIC COMMENTS

1. A more detailed description of the approaches in sections 2.3.1 and 2.3.2 is required. Particularly, the connections between Equations 8 to 14 need better explanation. Furthermore, the links between $R_{meas}$ and R, and between C, and $C_{ref}$, as well as their respective wavelength dependencies need to be introduced. As an example, the authors say on lines 243 ff that "By applying a linear fit to Eq. (8) vs. ATN data, it is possible to obtain the shadowing factor as follows …". However, the connection of Eq (8) and Eq (9) via the claimed fit procedure is not clear. Then, on lines 248 ff, C is parameterized as a function of AAE, although the physically based relationship is on the wavelength. Again, the approach for this parameterization is not clearly described.

In Eq. (11), the wavelength dependence of the single-scattering albedo (SSA) is parameterized as a function of the AAE, although the physical-based relationship is on the wavelength. Furthermore, the parameters required for a direct determination of SSA as a function of $\lambda$ are available: $\sigma_{sca}$ is measured for 3 wavelengths and $\sigma_{abs}$ is measured for one wavelength, so that SSA can be determined directly for one wavelength and extrapolated to the other wavelengths by the described iteration procedure. The direct approach would avoid the assumption that $\sigma_{sca}$ and the scattering contribution of the SSA scale both with $å_{sca}$ and same for the absorption part. A comment is requested why this approach was chosen.

2. The authors have excluded the comparison of $å_{atn}$ from AE data to $å_{abs}$ from MWAA data; see Fig. 6, although they state that the original attenuation Ångström exponent was also found to fit very well the MWAA-retrieved AAE, see Fig. S4. I recommend to include the $å_{atn}$ values in the intercomparison and to discuss whether or not the wavelength-dependence of $å_{abs}$ is affected by the correction algorithms. The potential result that the original attenuation Ångström exponents may give reliable estimates of the wavelength dependence of $\sigma_{abs}$ would be very important.

3. The discussion of the results presented in Fig. 5 is critical. Here, the authors compare absorption coefficients obtained from MAAP and MWAA and found significant differences. When inserting the 1:1 line it becomes evident that $\sigma_{abs}$ from MAAP are always larger than respective values from MWAA. Although the authors find a slope of 1.04 for the polluted period, there is a statistically

significant offset of 1.18 Mm$^{-1}$. In the considered range of $\sigma_{abs}$ values the offset can reach more than 10% of the total value. Furthermore, the data for the cleaner period with $\sigma_{abs} < 5$ Mm$^{-1}$ are highly correlated which conflicts the argument, that the under-determination of $\sigma_{abs}$ for low values may be explained by the proximity to the detection limit of the MWAA method. If this would be the case, I would expect a less linear relationship between the $\sigma_{abs}$ values with arbitrarily scattered values from the MWAA method. A careful discussion of the results shown in Fig. 5 is recommended.

4. A clear conclusion of the study is required. What have we learned from the presented work? Do the authors recommend the adapted Collaud Coen algorithm for future use, or one of the other two investigated approaches?

MINOR COMMENTS

1. The nomenclature used in the manuscript requires careful cross-checking, particularly for the following issues:

- BC mass concentrations obtained from light absorption methods are referred to as equivalent BC (eBC). This acronym should be used throughout the manuscript whenever appropriate.
- The absorption Ångström exponent is referred to as AAE, as $\mathring{a}_{ABS}$, or as $\mathring{a}_{abs}$.
- The light absorption coefficient is referred to as $\sigma_{abs}$, as $\sigma_{ABS}$, or as $\sigma_{ap}$. Consistency is requested.
- The city of Genoa is referred to as Genoa or Genova, please check.

2. The complete reference (Kirchstetter et al., 2004) should be listed in the bibliography.

3. When referring to Virkkula's PSAP correction scheme (Virkkula et al., 2005), also the correction (Virkkula, 2010) should be referenced.

4. Line 66: The sentence "retrieving the wavelength dependence of ambient aerosol requires …" is misleading. I assume the authors mean "retrieving the wavelength dependence of ambient aerosol optical properties requires …".

5. Line 117: The acronym PAS should be introduced.

6. Line 169: The correct name of the MAAP is (Model 5012, Thermo Electron Group, Waltham, USA).

7. Line 204: The correct equation deduced from the Lambert-Beer law is ATN = 100 ln ($I_0$ / I) or ATN = -100 ln (I/$I_0$); see e.g. (Hansen et al., 1982; Petzold et al., 1997; Weingartner et al., 2003).

8. Line 211: The meaning of „ (14625/λ)" is not clear. If it should refer to the parameterization of $\alpha_{ATN}$ values as a function of wavelength, a clarification and the addition of units are needed.

9. Eq. (6). Why not inserting the MAAP wavelength value given one line above instead of using the variable $\lambda_{MAAP}$?

10. Line 238: Please specify for which variable the parameter $C_{ref}$ was averaged.

11. The use of reference wavelengths of 637 nm or 880 nm is confusing; see Section 3.1. A short explanation may help to clarify why the specific wavelength is used as a reference.

12. Line 436: On a statistical basis, the offset of the Collaud Coen algorithm of -0.07 $\pm$ 0.30 is not slightly negative but indistinguishable from zero. This statement should be corrected.

13. Figures: For all scatter plots, 1:1 lines and grid lines should be shown as guidelines to the reader. In Fig. 7, the range of x- and y-axes should be similar, it might be of advantage to show only the relevant AAE - range from 0.5 to 2.0.

REFERENCES

Collaud Coen, M., Weingartner, E., Apituley, A., Ceburnis, D., Fierz-Schmidhauser, R., Flentje, H., Henzing, J. S., Jennings, S. G., Moerman, M., Petzold, A., Schmid, O., and Baltensperger, U.: Minimizing light absorption measurement artifacts of the Aethalometer: evaluation of five correction algorithms, Atmos. Meas. Tech., 3, 457-474, doi: 10.5194/amt-3-457-2010, 2010.

Hansen, A. D. A., Rosen, H., and Novakov, T.: Real-time measurement of the aerosol absoprtion-coefficient of aerosol particles, Appl. Opt., 21, 3060-3062, 1982.

Kirchstetter, T. W., Novakov, T., and Hobbs, P. V.: Evidence that the spectral dependence of light absorption by aerosols is affected by organic carbon, J. Geophys. Res., 109, D21208, doi: 10.1029/2004JD004999, 2004.

Massabò, D., Bernardoni, V., Bove, M. C., Brunengo, A., Cuccia, E., Piazzalunga, A., Prati, P., Valli, G., and Vecchi, R.: A multi-wavelength optical set-up for the characterization of carbonaceous particulate matter, J. Aerosol Sci., 60, 34-46, doi: http://dx.doi.org/10.1016/j.jaerosci.2013.02.006, 2013.

Massabò, D., Caponi, L., Bernardoni, V., Bove, M. C., Brotto, P., Calzolai, G., Cassola, F., Chiari, M., Fedi, M. E., Fermo, P., Giannoni, M., Lucarelli, F., Nava, S., Piazzalunga, A., Valli, G., Vecchi, R., and Prati, P.: Multi-wavelength optical determination of black and brown carbon in atmospheric aerosols, Atmos. Environ., 108, 1-12, doi: http://dx.doi.org/10.1016/j.atmosenv.2015.02.058, 2015.

Petzold, A., Kopp, C., and Niessner, R.: The dependence of the specific attenuation cross-section on black carbon mass fraction and particle size, Atmos. Environ., 31, 661-672, 1997.

Schmid, O., Artaxo, P., Arnott, W. P., Chand, D., Gatti, L. V., Frank, G. P., Hoffer, A., Schnaiter, M., and Andreae, M. O.: Spectral light absorption by ambient aerosols influenced by biomass burning in the Amazon Basin. I: Comparison and field calibration of absorption measurement techniques, Atmos. Chem. Phys., 6, 3443–3462, doi: 10.5194/acp-6-3443-2006, 2006.

Virkkula, A., Ahlquist, N. C., Covert, D. S., Arnott, W. P., Sheridan, P. J., Quinn, P. K., and Coffman, D. J.: Modification, calibration and a field test of an instrument for measuring light absorption by particles, Aerosol Sci. Technol., 39, 68-83, 2005.

Virkkula, A.: Correction of the Calibration of the 3-wavelength Particle Soot Absorption Photometer (3 PSAP), Aerosol Sci. Technol., 44, 706-712, doi: 10.1080/02786826.2010.482110, 2010.

Weingartner, E., Saathoff, H., Schnaiter, M., Streit, N., Bitnar, B., and Baltensperger, U.: Absorption of light by soot particles: determination of the absorption coefficient by means of aethalometers, J. Aerosol Sci., 34, 1445-1463, doi: 10.1016/S0021-8502(03)00359-8, 2003.

---

## Author Comment (AC1) · 3 Feb 2017

**Response to RC1**

We appreciate the referee comments and ideas that helped to improve the manuscript. Our responses are presented below (in black the original comments from the referee and our responses in green).

The paper presents an analysis of multiwavelength absorption data collected at an Amazonian site. Aerosol optical properties were measured with an aethalometer, MAAP and a nephelometer. The MAAP filter spots were later analyzed with an offline method, the MWAA that was considered here as the absorption standard. The MWAA is based on the same principle as the MAAP since it measures both transmitted and scattered light and a radiative transfer algorithm similar to that in the MAAP is applied to calculate the absorption coefficient. The good point is that it has several wavelengths, the weak points are that it is still a filter-based method with related artifacts and that its time resolution is not as good as that of online instruments.
The aethalometer data were processed with two methods, the Schmid et al. (2006) and the Collaud Coen et al. (2010) algorithm and the important Cref value was retrieved for both methods. The analysis shows that there are large sources of uncertainty of Cref and the wavelength dependency of absorption (AAE). One of the algorithms seems to be better in one respect, the other in another way. Absolute truth is still not found.

> Although absolute truth is not found, our study presents new evidence that helps to evaluate the efficiency of the different AE correction algorithms to retrieve the absorption wavelength dependence. The main advantage of this work is that AE corrected data was compared to a multi-wavelength measurement that is compensated for multiple scattering effects. Given that the filter loading effect was not significant, the uncertainties in the reference multi-wavelength method are minimal, although still a filter-based method.

1. My first question is related to Eq. (11). On line 256 you write that you use five different AAEs to calculate SSA and further Cref. Does this not result in five different Crefs? Do you give the average as the final Cref? Why not using the calculated AAEs instead of five fixed values? The whole procedure is not clear enough and unambiguously explained so that I would try to apply it to my own data.

> A reasonable range of AAE is used to calculate wavelength-dependent $C$ values, which are then used to fit $\ln(C)$ vs. $\ln(\lambda)$. The obtained fit parameters are fitted vs. AAE in order to parameterize $C$ as a function of $\lambda$ and AAE. Being AAE unknown in the beginning and giving the fact that C depends on AAE, the parameterization was the optimal approach Schmid proposed to solve this issue.

> The five different AAEs are only used for this parameterization and the calculated AAEs are used later in the algorithm to obtain the final absorption coefficients.

> Our scripts are available in the following link:

> https://dx.doi.org/10.6084/m9.figshare.c.3501153.v3

> In order to make the procedure clearer we modified the manuscript as follows:

> **Changes to Section 2.3.1:**

> "Attenuation coefficients at 590 nm were interpolated to 637 nm assuming a power-law relationship as,

$$\sigma_{\text{ATN}}(637\,\text{nm})=\sigma_{\text{ATN}}(590\,\text{nm})\cdot\left(\frac{637\,\text{nm}}{590\,\text{nm}}\right)^{-\mathring{a}_{\text{ATN}}} \qquad (7)\text{"}$$

"The compensated absorption coefficients, $\sigma_{\text{ap}}$, are calculated from attenuation coefficients, $\sigma_{\text{ATN}}$, by accounting for the different artifacts,

$$\sigma_{\text{ap}}=\frac{\sigma_{\text{ATN}}}{\left(C_{\text{ref}}+C_{\text{sca}}\right)\cdot R}$$

$$=\frac{\sigma_{\text{ATN}}}{\left(C_{\text{ref}}+m_{\text{s}}\dfrac{\omega_0}{1-\omega_0}\right)\left[\left(\dfrac{1}{f}-1\right)\left(\dfrac{\ln\text{ATN}-\ln 10}{\ln 50-\ln 10}\right)+1\right]} \qquad (5)$$

where $C_{\text{ref}}$ compensates for the scattering effects in comparison with a reference absorption measurement, $C_{\text{sca}}$ accounts for the scattering effect of non-absorbing aerosol particles and $R$, for the filter-loading effect. The Schmid formulation uses the scattering factor $m_{\text{s}}$ and $\omega_0$ to calculate $C_{\text{sca}}$ and the filter loading correction proposed by Weingartner et al. (2003), which takes ATN = 10 % as a reference point and includes the shadowing factor parameter, $f$, which describes the slope between $\sigma_{\text{ATN}}$ and ln(ATN)."

"The slope of this relationship was given by the shadowing factor parameter, $f$. By applying a linear fit to the $R_{\text{meas}}$ values obtained from Eq. (9) and the attenuation data, as shown in Eq. (10), the term $(1/f-1)$ can be obtained from the slope.

$$R_{\text{meas}}=\left(\frac{1}{f}-1\right)\left(\frac{\ln\text{ATN}-\ln 10}{\ln 50-\ln 10}\right)+1 \qquad (10)$$

Assuming $f$ is wavelength independent, the averaged $f$ is the used to calculate $R$ at different wavelengths."

"In the next step, $C$, understood as the overall multiple scattering correction factor ($C_{\text{ref}}+C_{\text{sca}}$), is parameterized as a function of $\lambda$. The single scattering albedo, $\omega_0$, at 637 nm is used in the following equation to calculate $C$ as

$$C=C^{*}+m_{\text{s}}\frac{\omega_0}{1-\omega_0} \qquad (11)$$

where $C^{*}$ corresponds to the multiple scattering effect by filter fibers and $m_{\text{s}}$ to the aerosol scattering factor found by Arnott et al. (2005) (see Table S1). The implemented approach is useful to examine any wavelength dependence on $C$. The values of $\omega_0$ are interpolated to the different Aethalometer wavelengths by using the Eq. (12), assuming that absorption and scattering coefficients follow a power-law wavelength dependence described by $\mathring{a}_{\text{ABS}}$ and $\mathring{a}_{\text{SCA}}$, respectively.

$$\omega_0(\lambda)=\frac{\sigma_{\text{sp}}}{\sigma_{\text{sp}}+\sigma_{\text{ap}}}$$

$$=\frac{\omega_{0,\text{ref}}\left(\dfrac{\lambda}{\lambda_{\text{ref}}}\right)^{-\mathring{a}_{\text{SCA}}}}{\omega_{0,\text{ref}}\left(\dfrac{\lambda}{\lambda_{\text{ref}}}\right)^{-\mathring{a}_{\text{SCA}}}+\left(1-\omega_{0,\text{ref}}\right)\left(\dfrac{\lambda}{\lambda_{\text{ref}}}\right)^{-\mathring{a}_{\text{ABS}}}} \qquad (12)$$

Different $\mathring{a}_{\text{ABS}}$ values (1; 1.25; 1.5; 1.75; 2) are then used to generate different correlation factors between $\ln(C)$ vs. $\ln(\lambda)$. The coefficients resulting from a quadratic fit are used to parameterize $C$ as a function of $\mathring{a}_{\text{ABS}}$ (see Fig. 4 in Schmid et al. (2006)). An iteration procedure is used to force the convergence of $\mathring{a}_{\text{ABS}}$. In our calculations, the data converged after seven iterations."

**Changes to section 2.3.2:**
"In this study we implemented the Collaud Coen correction algorithm that resembles the

Schmid correction (see eq. 14b in Collaud Coen et al. (2010)). This algorithm is different from the original Schmid algorithm in the calculations of the filter-loading effect and the multiple scattering correction factor. As shown in Eq. (6), the Schmid algorithm filters the data for ATN < 10 % in order to account only for the scattering by filter fibers in the $C_{ref}$ calculation. On the other hand, Collaud Coen algorithm applies a prior filter-loading correction and then, by dividing the reference absorption data (MAAP) by the Aethalometer attenuation coefficients, they obtain $C_{ref}$, which accounts for both, scattering by filter fibers and scattering by embedded aerosol particles."

"Finally, the corrected absorption coefficients are calculated in a similar way to Eq. (5) but using $m_s$ from Eq. (15) and averaging $C_{ref}$, $m_s$, $\omega_0$ and $R$ over a filter spot period; i.e., from a filter change time to the subsequent one."

**Section 2.3.3 was merged with 2.3.2.**

2. Line 256. "Using different AAE (å_abs = 1,..." Why do you use the symbols AAE and å_abs for the same thing? Be consistent throughout the text.

All "AAE" acronyms were replaced by the symbol "$å_{ABS}$".

3. L384-389 "A scatter plot of both corrections' outputs vs. MAAP measurements is shown in Fig. 4. We found that corrected AE data fitted very well the MAAP measurements in the case of the Schmid correction, with a slope of 1.04 (1.02 – 1.05); i.e, the Schmid correction overestimates the absorption coefficient by only 2– 5 %. In the case of the Collaud Coen correction, it was found that the AE corrected absorption coefficients were underestimated by 19 – 21 %." I don't understand. My first thought is that the Crefs are just wrong. As far as I have understood the whole thing of getting Cref is based on forcing the aethalometer-derived absorption to match the MAAP-derived absorption. Should not the slope should be very close to if not exactly 1 regardless of the algorithm selected? Please explain.

We found an error in our algorithms that affected the $C_{sca}$ term of the Collaud Coen correction and scaled down the absorption coefficient obtained with this correction. After fixing the error, we updated Fig. 4, Fig 3c, and Fig. 8. Now both corrections show a good comparison to the MAAP at 637 nm. The update in the algorithms did not affect the wavelength dependence of the absorption. All updated figures and discussion are attached to this document.

4. L439-440. It is reminded to evaluate critically the corrected filter-based absorption data when using it to retrieve BrC / BC contributions. There are quite a few papers that discuss this same issue. Give some references to such papers, some more credit to earlier work could be given.

We rephrased as:
"A near-UV over- or underestimation of the data, will substantially affect brown carbon calculations, if apportionment algorithms based on the wavelength dependence of absorption are used. More details on the effects of inaccurate $å_{ABS}$ on the BrC/BC apportionment are discussed in Garg et al., 2016; Schuster et al., 2016a, 2016b; Wang et al., 2016 and references there in. A BrC estimation is beyond the scope of this paper."
The included references are detailed below.

5. Fig 3. Beta should have units. scattering coefficient = beta x lambdaˆ(-alpha)

> The referee is right. We added the units to Fig. 3 (a).

6. Figures 6 and S4 are just the same with the exception that in S4 there is one more line. Why would you not show it simply in Fig 6 and omit S4?

> We agree. Figure 6 was updated to include all data included in Fig. S4. Figure S4 was removed from the supplementary material.

*# End of referee comments and author responses #*

**References**

[revised manuscript text omitted]

---

## Author Comment (AC3) · 8 Feb 2017

[revised manuscript text omitted]
 \dfrac{\omega_0}{1-\omega_0}\right)\left[\left(\dfrac{1}{f}-1\right)\left(\dfrac{\ln ATN - \ln 10}{\ln 50 - \ln 10}\right)+1\right]} \tag{5}$$

where $C_{ref}$ compensates for the scattering effects in comparison with a reference absorption measurement, $C_{sca}$ accounts for the scattering effect of non-absorbing aerosol particles and $R$, for the filter-loading effect. The Schmid formulation uses the scattering factor $m_s$ and $\omega_0$ to calculate $C_{sca}$ and the filter loading correction proposed by Weingartner et al. (2003), which takes ATN = 10 % as a

295 reference point and includes the shadowing factor parameter, $f$, which describes the slope between $\sigma_{ATN}$ and ln(ATN).

As a first step, $C_{ref}$ is calculated for attenuation coefficients corresponding to attenuation values lower than 10 %, when the filter-loading correction is considered negligible (ATN < 10 %; R ≈ 1). By using MAAP absorption coefficient measurements, it is possible to obtain $C_{ref}$ as follows:

300

attenuation coefficients corresponding to attenuation values lower than 10 %, when the filter-loading correction is considered negligible (ATN < 10 %; R ≈ 1). By using MAAP absorption coefficient measurements, it is possible to obtain $C_{ref}$ as follows:

305 $$C_{ref} = \frac{\sigma_{ATN,10}}{\sigma_{MAAP}} \qquad (65)$$

where $\sigma_{MAAP}$ is the absorption coefficient measured by the MAAP at 637 nm and $\sigma_{ATN, 10}$ is the attenuation coefficient at 637 nm ($\lambda_{MAAP}$) when ATN < 10 %.

Attenuation coefficients at 590 nm were interpolated to 637 nm assuming a power-law relationship as,

$$\sigma_{ATN}(637\,nm) = \sigma_{ATN}(590\,nm) \cdot \left(\frac{637\,nm}{590\,nm}\right)^{-\mathring{a}_{ATN}} \qquad (76)$$

310 The attenuation Ångström exponent $\mathring{a}_{ATN}$ used in this step was calculated by applying a log-log fit to $\sigma_{ATN}$ vs. $\lambda$, where $\mathring{a}_{ATN}$ was obtained from the slope as follows:

$$\ln \sigma_{ATN} = -\mathring{a}_{ATN} \ln(\lambda) + \ln(constant) \qquad (87)$$

Absorption Ångström exponents ($\mathring{a}_{ABS}$) were obtained in a similar way in further calculations.

The multiple scattering correction factor, $C_{ref}$, obtained from Eq. (65) was averaged over the sampling

315 period to calculate the measured filter-loading correction factor, $R_{meas}$, as

$$\text{————————} \quad (8)$$

Weingartner et al. (2003) found that the linear relationship between $\sigma_{ATN}$ and ln(ATN) can be used to parameterize the filter-loading effect. The slope of this relationship was given by a parameter called the *shadowing factor*, *f*. By applying a linear fit to Eq. (8) vs. ATN data, it is possible to obtain the

320 shadowing factor as follows

$$R_{meas} = \frac{\sigma_{ATN}}{\sigma_{MAAP} \cdot \overline{C}_{ref}} \qquad (9)$$

Weingartner et al. (2003) found that the linear relationship between $\sigma_{ATN}$ and ln(ATN) can be used to parameterize the filter-loading effect. The slope of this relationship was given by the shadowing factor parameter, *f*. By applying a linear fit to the $R_{meas}$ values obtained from Eq. (9) and the attenuation data,

325 as shown in Eq. (10), the term ($1/f - 1$) can be obtained from the slope.

Assuming *f* is wavelength independent, the averaged *f* is used to calculate *R* from Eq. (9) at different wavelengths.

The next step is the parameterization of *C* as a function of the AAE. In order to do that, *C* was rewritten as:

$$R = \left(\frac{1}{f} - 1\right)\left(\frac{\ln \text{ATN} - \ln 10}{\ln 50 - \ln 10}\right) + 1 \qquad (10)$$

Assuming *f* is wavelength independent, the averaged *f* is the used to calculate *R* at different wavelengths.

In the next step, *C*, understood as the overall multiple scattering correction factor ($C_{\text{ref}} + C_{\text{sca}}$), is parameterized as a function of λ. The single scattering albedo, $\omega_0$, at 637 nm is used in the following equation to calculate *C* as

where *C\** corresponds to the multiple scattering effect by filter fibers, $m_s$ to the aerosol scattering effect and $\omega_0$ to the single scattering albedo. The *C\** and $m_s$ values were taken from Arnott et al. (2005) (see Table S1), and the $\omega_0$ was calculated at 637 nm using measured scattering and absorption coefficients at 637 nm. The values of $\omega_0$ at different AE wavelengths were obtained by using the following equation:

$$C = C^* + m_s \frac{\omega_0}{1 - \omega_0} \qquad (11)$$

where *C\** corresponds to the multiple scattering effect by filter fibers and $m_s$ to the aerosol scattering factor found by Arnott et al. (2005) (see Table S1). The implemented approach is useful to examine any wavelength dependence on *C*. The values of $\omega_0$ were interpolated to the different Aethalometer wavelengths by using the Eq. (12), assuming that absorption and scattering coefficients follow a power-law wavelength dependence described by $\mathring{a}_{\text{ABS}}$ and $\mathring{a}_{\text{SCA}}$, respectively.

$$\omega_0(\lambda) = \frac{\sigma_{\text{sp}}}{\sigma_{\text{sp}} + \sigma_{\text{ap}}}$$

$$= \frac{\omega_{0,\text{ref}}\left(\frac{\lambda}{\lambda_{\text{ref}}}\right)^{-\mathring{a}_{\text{SCA}}}}{\omega_{0,\text{ref}}\left(\frac{\lambda}{\lambda_{\text{ref}}}\right)^{-\mathring{a}_{\text{SCA}}} + (1 - \omega_{0,\text{ref}})\left(\frac{\lambda}{\lambda_{\text{ref}}}\right)^{-\mathring{a}_{\text{ABS}}}} \qquad (12)$$

[revised manuscript text omitted]

---

## Author Comment (AC2)

**Response to RC2**

The authors thank the referee for the pertinent comments and the ideas to improve the manuscript. Our responses are presented below (in black the original comments from the referee and our responses in green).

GENERAL COMMENT

The study targets the measurement of absorption Ångström exponents (AAE) by filter-based multi-wavelength light absorption measurement methods. Light attenuation data from the widely used 7-wavelength Aethalometer are analyzed by applying two correction schemes according to Schmid et al. (2006) and Collaud Coen et al. (2010) and comparing the results to data from the filter-based offline Multi-Wavelength Absorbance Analyzer MWAA (Massabò et al., 2013; Massabò et al., 2015). Reference methods of the measurement of light scattering and light absorption were a 3-λ Integrating Nephelometer (Model Aurora 3000, Ecotech) and a single-wavelength Multi-Angle Absorption Photometer MAAP (Model 5012, Thermo Electron Group). Data have been collected at the ATTO site in Amazonia from the wet-to-dry transition season to the dry season in 2014.

The topic of the study is of relevance for the research area of determining black carbon (BC) and brown carbon (BrC) from the wavelength dependence of the light absorption coefficient, characterized by the AAE. The presented data is well suited for the study and of high importance and thus deserves publication in AMT. However, before being suitable for publication, the manuscript requires major revisions which are highlighted in the following.

> We agree with most of the comments presented by the referee. After a major revision of the manuscript, we consider we have addressed all of the referee concerns.

SPECIFIC COMMENTS

1. A more detailed description of the approaches in sections 2.3.1 and 2.3.2 is required. Particularly, the connections between Equations 8 to 14 need better explanation. Furthermore, the links between $R_{meas}$ and R, and between C, and $C_{ref}$, as well as their respective wavelength dependencies need to be introduced. As an example, the authors say on lines 243 ff that "By applying a linear fit to Eq. (8) vs. ATN data, it is possible to obtain the shadowing factor as follows ...". However, the connection of Eq (8) and Eq (9) via the claimed fit procedure is not clear. Then, on lines 248 ff, C is parameterized as a function of AAE, although the physically based relationship is on the wavelength. Again, the approach for this parameterization is not clearly described.

In Eq. (11), the wavelength dependence of the single-scattering albedo (SSA) is parameterized as a function of the AAE, although the physical-based relationship is on the wavelength. Furthermore, the parameters required for a direct determination of SSA as a function of λ are available: $\sigma_{sca}$ is measured for 3 wavelengths and $\sigma_{abs}$ is measured for one wavelength, so that SSA can be determined directly for one wavelength and extrapolated to the other wavelengths by the described iteration procedure. The direct approach would avoid the assumption that $\sigma_{sca}$ and the scattering contribution of the SSA scale both with $å_{sca}$ and same for the absorption part. A comment is requested why this approach was chosen.

> We changed sections 2.3.1 and 2.3.2 to make the procedure clearer to the reader and fulfill the referee requests. Regarding the SSA parameterization expressed in Eq. (11) (now Eq. (12)), we used the presented approach to study the wavelength dependence on the multiple-scattering correction.

**Changes to Section 2.3.1:**
"Attenuation coefficients at 590 nm were interpolated to 637 nm assuming a power-law relationship as,

$$\sigma_{ATN}(637\,\mathrm{nm}) = \sigma_{ATN}(590\,\mathrm{nm}) \cdot \left(\frac{637\,\mathrm{nm}}{590\,\mathrm{nm}}\right)^{-\mathring{a}_{ATN}} \qquad (7)"$$

"The compensated absorption coefficients, $\sigma_{ap}$, are calculated from attenuation coefficients, $\sigma_{ATN}$, by accounting for the different artifacts,

$$\sigma_{ap} = \frac{\sigma_{ATN}}{(C_{ref} + C_{sca}) \cdot R}$$
$$= \frac{\sigma_{ATN}}{\left(C_{ref} + m_s \dfrac{\omega_0}{1-\omega_0}\right)\left[\left(\dfrac{1}{f}-1\right)\left(\dfrac{\ln ATN - \ln 10}{\ln 50 - \ln 10}\right)+1\right]} \qquad (5)$$

where $C_{ref}$ compensates for the scattering effects in comparison with a reference absorption measurement, $C_{sca}$ accounts for the scattering effect of non-absorbing aerosol particles and $R$, for the filter-loading effect. The Schmid formulation uses the scattering factor $m_s$ and $\omega_0$ to calculate $C_{sca}$ and the filter loading correction proposed by Weingartner et al. (2003), which takes ATN = 10 % as a reference point and includes the shadowing factor parameter, $f$, which describes the slope between $\sigma_{ATN}$ and ln(ATN)."

"The slope of this relationship was given by the shadowing factor parameter, $f$. By applying a linear fit to the $R_{meas}$ values obtained from Eq. (9) and the attenuation data, as shown in Eq. (10), the term $(1/f-1)$ can be obtained from the slope.

$$R_{meas} = \left(\frac{1}{f}-1\right)\left(\frac{\ln ATN - \ln 10}{\ln 50 - \ln 10}\right)+1 \qquad (10)$$

Assuming $f$ is wavelength independent, the averaged $f$ is the used to calculate $R$ at different wavelengths."

"In the next step, $C$, understood as the overall multiple scattering correction factor ($C_{ref}$ + $C_{sca}$), is parameterized as a function of λ. The single scattering albedo, $\omega_0$, at 637 nm is used in the following equation to calculate $C$ as

$$C = C^* + m_s \frac{\omega_0}{1-\omega_0} \qquad (11)$$

where $C^*$ corresponds to the multiple scattering effect by filter fibers and $m_s$ to the aerosol scattering factor found by Arnott et al. (2005) (see Table S1). The implemented approach is useful to examine any wavelength dependence on $C$. The values of $\omega_0$ are interpolated to the different Aethalometer wavelengths by using the Eq. (12), assuming that absorption and scattering coefficients follow a power-law wavelength dependence described by $\mathring{a}_{ABS}$ and $\mathring{a}_{SCA}$, respectively.

$$\omega_0(\lambda) = \frac{\sigma_{sp}}{\sigma_{sp} + \sigma_{ap}}$$
$$= \frac{\omega_{0,ref}\left(\dfrac{\lambda}{\lambda_{ref}}\right)^{-\mathring{a}_{SCA}}}{\omega_{0,ref}\left(\dfrac{\lambda}{\lambda_{ref}}\right)^{-\mathring{a}_{SCA}} + (1-\omega_{0,ref})\left(\dfrac{\lambda}{\lambda_{ref}}\right)^{-\mathring{a}_{ABS}}} \qquad (12)$$

Different $\mathring{a}_{ABS}$ values (1; 1.25; 1.5; 1.75; 2) are then used to generate different correlation factors between ln($C$) vs. ln(λ). The coefficients resulting from a quadratic fit are used to parameterize $C$ as a function of $\mathring{a}_{ABS}$ (see Fig. 4 in Schmid et al. (2006)). An iteration procedure is used to force the convergence of $\mathring{a}_{ABS}$. In our calculations, the data converged after seven iterations."

**Changes to section 2.3.2:**
"In this study we implemented the Collaud Coen correction algorithm that resembles the Schmid correction (see eq. 14b in Collaud Coen et al. (2010)). This algorithm is different from the original Schmid algorithm in the calculations of the filter-loading effect and the multiple scattering correction factor. As shown in Eq. (6), the Schmid algorithm filters the data for ATN < 10 % in order to account only for the scattering by filter fibers in the $C_{ref}$ calculation. On the other hand, Collaud Coen algorithm applies a prior filter-loading correction and then, by dividing the reference absorption data (MAAP) by the Aethalometer attenuation coefficients, they obtain $C_{ref}$, which accounts for both, scattering by filter fibers and scattering by embedded aerosol particles."

"Finally, the corrected absorption coefficients are calculated in a similar way to Eq. (5) but using $m_s$ from Eq. (15) and averaging $C_{ref}$, $m_s$, $\omega_0$ and $R$ over a filter spot period; i.e., from a filter change time to the subsequent one."

**Section 2.3.3 was merged with 2.3.2.**

2. The authors have excluded the comparison of $å_{atn}$ from AE data to $å_{abs}$ from MWAA data; see Fig. 6, although they state that the original attenuation Ångström exponent was also found to fit very well the MWAA-retrieved AAE, see Fig. S4. I recommend to include the $å_{atn}$ values in the intercomparison and to discuss whether or not the wavelength-dependence of $å_{abs}$ is affected by the correction algorithms. The potential result that the original attenuation Ångström exponents may give reliable estimates of the wavelength dependence of $\sigma_{abs}$ would be very important.

In the new version of the manuscript we give more relevance and discuss more about the comparison between the MWAA $å_{ABS}$ and $å_{ATN}$. The $å_{ATN}$ data shown in Fig. S4 is now added to Fig. 6.

**New figure:**

[Figure]

We added the following text to the manuscript:

**Section 3.1**
"The original attenuation Ångström exponent (without applying any compensation) was also found to fit quite well the MWAA-retrieved $å_{ABS}$, (Slope IQR: 0.89 – 1.10 with R² = 0.75,

not shown). This finding is in accordance with Ajtai et al., 2011 who found a good agreement between 4-λ PAS measurements and the Aethalometer raw wavelength dependence at a sub-urban site."

**Conclusions**
"On the other hand, the Collaud Coen algorithm as well as the "raw" Aethalometer attenuation spectral dependence reproduced quite well the $\text{å}_{\text{ABS}}$ values obtained from MWAA measurements."

**Abstract**
"Additionally, we found that the wavelength dependence of uncompensated Aethalometer attenuation data significantly correlates with the $\text{å}_{\text{ABS}}$ retrieved from offline MWAA measurements."

3. The discussion of the results presented in Fig. 5 is critical. Here, the authors compare absorption coefficients obtained from MAAP and MWAA and found significant differences. When inserting the 1:1 line it becomes evident that $\sigma_{\text{abs}}$ from MAAP are always larger than respective values from MWAA. Although the authors find a slope of 1.04 for the polluted period, there is a statistically significant offset of 1.18 Mm$^{-1}$. In the considered range of $\sigma_{\text{abs}}$ values the offset can reach more than 10% of the total value. Furthermore, the data for the cleaner period with $\sigma_{\text{abs}} < 5$ Mm$^{-1}$ are highly correlated which conflicts the argument, that the under-determination of $\sigma_{\text{abs}}$ for low values may be explained by the proximity to the detection limit of the MWAA method. If this would be the case, I would expect a less linear relationship between the $\sigma_{\text{abs}}$ values with arbitrarily scattered values from the MWAA method. A careful discussion of the results shown in Fig. 5 is recommended.

We agree with the referee on his concerns about the interpretation of Fig. 5 and its implications. We have changed the discussion on section 3.2 as follows:

**Original text:**
The MWAA was used as a reference multi-wavelength measurement since it accounts for multiple scattering effects by means of a similar configuration to the MAAP. Light absorption coefficients obtained from the MWAA (at 635 nm) and from the MAAP (at 637 nm) were compared by applying an linear regression to both datasets after integrating the MAAP data over the filter total sampling times. The fit resulted in an MWAA underestimation by 14 to 18% when fitting the whole dataset. However, when comparing only data from the polluted period (18 – 23 August 2014), the MWAA underestimation was only ~5 %. A scatter plot, including the fits, can be seen in Fig. 5. The MWAA underestimation for low absorption coefficient samples might be related to the proximity to the instrument detection limits. The possibility of losing part of the BrC aerosol of medium volatility was also considered and all data with $\sigma_{\text{ap}} < 1$ Mm$^{-1}$ were considered with caution when making any interpretation in the further analysis.

**Replaced by:**
"The MWAA was used as a reference multi-wavelength measurement since it accounts for multiple scattering effects by means of a similar configuration to the MAAP. Light absorption coefficients obtained from the MWAA (at 635 nm) and from the MAAP (at 637 nm) were compared by applying an linear regression to both datasets after integrating the MAAP data over the filter total sampling times, as shown in Fig. 5. The fit resulted in a MWAA underestimation of 14 to 18% when fitting the whole dataset. In general, all values measured by the MWAA at 635 nm were below the MAAP measurements at 637 nm with a decreasing offset towards lower absorption coefficients. This could be associated to a

significant volatilization of the absorbing aerosol collected during the polluted period. The comparison Aethalometer – MWAA at different wavelengths was based on the assumption that these losses are wavelength-independent."

**Updated figure:**

[Figure]

4. A clear conclusion of the study is required. What have we learned from the presented work? Do the authors recommend the adapted Collaud Coen algorithm for future use, or one of the other two investigated approaches?

We agree with the referee and improved the conclusions and discussion of our results.

The conclusions were changed to:
"We applied two different correction algorithms to compensate for the various Aethalometer absorption measurement artifacts. The compensated data was compared to an offline multi-wavelength reference absorption measurement technique. This comparison allowed studying the effects of the correction schemes on the absorption at lower wavelengths and showed how this affects the $å_{ABS}$ retrieval. We found that both analyzed algorithms efficiently reproduce the reference MAAP absorption coefficients from Aethalometer data. However, the Schmid algorithm overestimates the $å_{ABS}$ compared to that obtained by the multiple wavelength measurement (MWAA). On the other hand, the Collaud Coen algorithm as well as the "raw" Aethalometer attenuation spectral dependence reproduced quite well the $å_{ABS}$ values obtained from MWAA measurements. The under- or overestimation of short-wavelength absorption coefficients by compensation algorithms is a factor that has to be considered when using corrected Aethalometer data to apportion the black and brown carbon contributions to total absorption. When comparing the absorption coefficients obtained from the different correction algorithms to the reference measurement at 370 nm, we found that the Collaud Coen algorithm is more appropriate to achieve the best comparison at this wavelength, especially for data with $\sigma_{ap} > 5$ Mm$^{-1}$. The Schmid algorithm resulted in high enhancements of the absorption coefficients at 370 nm over the sampling period."

MINOR COMMENTS

1. The nomenclature used in the manuscript requires careful cross-checking, particularly for the following issues:

- BC mass concentrations obtained from light absorption methods are referred to as equivalent BC (eBC). This acronym should be used throughout the manuscript whenever appropriate.
  We agree but prefer to use the most commonly used acronym $BC_e$, instead of eBC.

- The absorption Ångström exponent is referred to as AAE, as $å_{ABS}$, or as $å_{abs}$.
  We homogenized the manuscript to use always $å_{ABS}$.

- The light absorption coefficient is referred to as $\sigma_{abs}$, as $\sigma_{ABS}$, or as $\sigma_{ap}$. Consistency is requested.
  Light absorption coefficient is now represented as $\sigma_{ap}$ throughout the text.

- The city of Genoa is referred to as Genoa or Genova, please check.
  All changed to "Genoa".

2. The complete reference (Kirchstetter et al., 2004) should be listed in the bibliography.
  Corrected.

3. When referring to Virkkula's PSAP correction scheme (Virkkula et al., 2005), also the correction (Virkkula, 2010) should be referenced.
  The reference to the correction has been included in the new version.

4. Line 66: The sentence "retrieving the wavelength dependence of ambient aerosol requires ..." is misleading. I assume the authors mean "retrieving the wavelength dependence of ambient aerosol optical properties requires ...".
  Corrected.

5. Line 117: The acronym PAS should be introduced.
  Corrected.

6. Line 169: The correct name of the MAAP is (Model 5012, Thermo Electron Group, Waltham, USA).
  Corrected.

7. Line 204: The correct equation deduced from the Lambert-Beer law is ATN = 100 ln ($I_0$ / I) or ATN =-100 ln ($I/I_0$); see e.g. (Hansen et al., 1982; Petzold et al., 1997; Weingartner et al., 2003).
  Corrected.

8. Line 211: The meaning of „ (14625/λ)" is not clear. If it should refer to the parameterization of $\alpha_{ATN}$ values as a function of wavelength, a clarification and the addition of units are needed.
  We agree. Changed to: "$\alpha_{ATN}$ is the $\lambda$-dependent BC mass attenuation cross-section (14625 nm m² g$^{-1}$ $\lambda^{-1}$)".

9. Eq. (6). Why not inserting the MAAP wavelength value given one line above instead of using the variable $\lambda_{MAAP}$?
  We agree. Changed to "637 nm".

10. Line 238: Please specify for which variable the parameter $C_{ref}$ was averaged.

In this statement we refer to $C_{ref}$ obtained from Eq. (5); i.e., the average is calculated using attenuation interpolated to 637 nm and absorption data from the MAAP. In order to clarify we added "averaged over the sampling period".

11. The use of reference wavelengths of 637 nm or 880 nm is confusing; see Section 3.1. A short explanation may help to clarify why the specific wavelength is used as a reference.

We agree that using the 880 nm wavelength as a reference might be confusing. We decided to report R values at 660 nm, which is the Aethalometer wavelength that is closer to 637 nm. Moreover, when mentioning the wavelength dependence of R we now calculate it for the full Aethalometer spectral range (370 – 960 nm).

The manuscript was changed to:
"At 660 nm, the Aethalometer wavelength that is closer to the MAAP measurement wavelength, the filter-loading correction calculation resulted in R correction factors of 0.98 ± 0.02 and 1.01 ± 0.01 for June – July and August – September, respectively. A slight wavelength dependence was observed; the *R* values were up to 4% higher at 370 nm compared to those calculated at 960 nm during the cleanest period of this study (June – July)."

12. Line 436: On a statistical basis, the offset of the Collaud Coen algorithm of -0.07 − 0.30 is not slightly negative but indistinguishable from zero. This statement should be corrected.

We agree. Discussion has been updated after an update to Fig. 8. Details are included in our response to referee #1.

13. Figures: For all scatter plots, 1:1 lines and grid lines should be shown as guidelines to the reader. In Fig. 7, the range of x- and y-axes should be similar, it might be of advantage to show only the relevant AAE - range from 0.5 to 2.0.

Fig. 7 has been updated and the new y-axis range goes from 0.0 to 2.0. We included 1:1 lines in all scatter plots but Fig. 4.

*# End of referee comments and author responses #*

**References**

Ajtai, T., Filep, Á., Utry, N., Schnaiter, M., Linke, C., Bozóki, Z., Szabó, G. and Leisner, T.: Inter-comparison of optical absorption coefficients of atmospheric aerosols determined by a multi-wavelength photoacoustic spectrometer and an Aethalometer under sub-urban wintry conditions, J. Aerosol Sci., 42(12), 859–866, doi:10.1016/j.jaerosci.2011.07.008, 2011.